# Intrinsic Lorentz Neural Network

**Xianglong Shi**[1]\*, **Ziheng Chen**[2]\*†, **Yunhan Jiang**[3], **Nicu Sebe**[2]
[1]University of Science and Technology of China, [2]University of Trento, [3]Peking University
xlshi@mail.ustc.edu.cn, ziheng_ch@163.com

## Abstract

Real-world data frequently exhibit latent hierarchical structures, which can be naturally represented by hyperbolic geometry. Although recent hyperbolic neural networks have demonstrated promising results, many existing architectures remain partially intrinsic, mixing Euclidean operations with hyperbolic ones or relying on extrinsic parameterizations. To address it, we propose the *Intrinsic Lorentz Neural Network* (ILNN), a fully intrinsic hyperbolic architecture that conducts all computations within the Lorentz model. At its core, the network introduces a novel *point-to-hyperplane* fully connected layer (FC), replacing traditional Euclidean affine logits with closed-form hyperbolic distances from features to learned Lorentz hyperplanes, thereby ensuring that the resulting geometric decision functions respect the inherent curvature. Around this fundamental layer, we design intrinsic modules: GyroLBN, a Lorentz batch normalization that couples gyro-centering with gyro-scaling, consistently outperforming both LBN and GyroBN while reducing training time. We additionally proposed a gyro-additive bias for the FC output, a Lorentz patch-concatenation operator that aligns the expected log-radius across feature blocks via a digamma-based scale, and a Lorentz dropout layer. Extensive experiments conducted on CIFAR-10/100 and two genomic benchmarks (TEB and GUE) illustrate that ILNN achieves state-of-the-art performance and computational cost among hyperbolic models and consistently surpasses strong Euclidean baselines. The code is available at this url.

## 1 Introduction

Hierarchical structures exist across a wide range of machine learning applications, including computer vision (Khrulkov et al., 2020; Ghadimi Atigh et al., 2022; Bdeir et al., 2024; Pal et al., 2025; Shi et al., 2025; Wang et al., 2026), natural language processing (Ganea et al., 2018; Tifrea et al., 2019; Yang et al., 2024a), knowledge-graph reasoning (Nickel & Kiela, 2017; Balazevic et al., 2019; Welz et al., 2025), graph learning (Chami et al., 2019; Yang et al., 2022; Li et al., 2024), brain signal decoding (Li et al., 2026), and genomics tasks (Khan et al., 2025; Zhou & Sharpee, 2021; Yang et al., 2025). While Euclidean neural networks often incur high distortion or require excessive dimensionality when embedding hierarchical or scale-free structures (Nickel & Kiela, 2018), hyperbolic geometry, characterized by constant negative curvature, offers exponential representational capacity and enables more compact embeddings (Ganea et al., 2018; Shimizu et al., 2020). Consequently, *hyperbolic neural networks* (HNNs) have demonstrated notable success in applications with hierarchical structure (Ganea et al., 2018; Chami et al., 2019; Shimizu et al., 2020; Gulcehre et al., 2019; Li et al., 2024; Fan et al., 2024; Leng et al., 2024; Bdeir et al., 2024; Pal et al., 2025; Chen et al., 2025a; He et al., 2025).

Early hyperbolic neural networks predominantly operated in the Poincaré model because its gyrovector formalism makes many neural primitives straightforward to define (Ganea et al., 2018; Shimizu et al., 2020). However, the unit-ball constraint and boundary saturation render the Poincaré ball more susceptible to numerical instabilities than the Lorentz model, motivating a recent shift toward Lorentz neural networks (LNNs) that exhibit improved optimization stability (Bdeir et al., 2024). Despite this trend leading to the emergence of excellent work (He et al., 2024; Fan et al., 2024; Liang et al., 2024), recent advanced LNNs (Bdeir et al., 2024; Khan et al., 2025) architectures remain only partially

---

*Equal contribution.
†Corresponding author.

intrinsic: they mix Euclidean affine transformations with manifold operations or rely on extrinsic parameterizations that compromise geometric consistency. For example, the Lorentz fully connected layer (LFC) (Bdeir et al., 2024) that applies Euclidean mappings to Lorentz vector, then extracts the space part from mapping output and accordingly calculates the time part to form the output Lorentz vector. Normalization further illustrates this tension. Lorentz batch normalization (LBN) (Bdeir et al., 2024) recenter features but either ignore gyro-variance, while GyroBN (Chen et al., 2025a;c) offers gyrogroup-based control yet can still be computationally heavy due to depending on Fréchet-type statistics. These compromises limit the representational power and efficiency of Lorentz neural networks.

An effective way to mitigate partially intrinsic designs is the *point-to-hyperplane* formulation, which has shown success in spherical, hyperbolic, and matrix geometries (Lebanon & Lafferty, 2004; Shimizu et al., 2020; Bdeir et al., 2024; Chen et al., 2024a;b; Nguyen et al., 2025). Inspired by this, we introduce ILNN, a fully intrinsic hyperbolic network in which every operation, parameter, and update is defined inside the Lorentz model. At its core is a *Point-to-hyperplane Lorentz Fully Connected* (PLFC) layer that replaces Euclidean affine transformation with closed-form Lorentzian distances to learned hyperplanes, yielding curvature-aware, margin-interpretable decision functions. Surrounding PLFC, we develop intrinsic components: *GyroLBN*, a batch normalization that couples gyro-centering with variance-controlled gyro-scaling and outperforms LBN and GyroBN while reducing wall-clock time. Furthermore, we introduced a log-radius algin Lorentz patch-concatenation to build the stable CNN module, a gyro-additive bias, and a Lorentz dropout layer to further improve its performance. By designing intrinsic geometric operations, ILNN maintains mathematical consistency and geometric interpretability throughout the network.

The main contributions of our work can be summarized as follows:

1. We propose an *intrinsic hyperbolic neural network* that eliminates reliance on extrinsic Euclidean operations, thus fully harnessing hyperbolic geometry.
2. We introduce a novel *point-to-hyperplane Lorentz fully connected layer*(PLFC), which replaces traditional affine transformations with intrinsic hyperbolic distances, significantly enhancing representational fidelity.
3. We introduce a GyroLBN, a Lorentz batch normalization that combines gyro-centering with gyro-scaling, consistently outperforming both LBN and GyroBN while reducing training time.
4. Extensive experiments demonstrate the effectiveness of ILNN, achieving state-of-the-art performance on CIFAR-10/100 datasets and genomic benchmarks (TEB and GUE), consistently outperforming Euclidean and existing hyperbolic counterparts.

## 2 RELATED WORK

**Hyperbolic embeddings.** A large body of work demonstrates that negatively curved representations can efficiently encode hierarchical and scale-free structure across modalities. Foundational results on hyperbolic embeddings (e.g., Poincaré embeddings) show strong benefits for symbolic data with latent trees (Nickel & Kiela, 2017), and early hyperbolic neural architectures extend core layers and classifiers to the Poincaré model for language tasks (Ganea et al., 2018). On graphs, hyperbolic GNNs capture hierarchical neighborhoods and outperform Euclidean counterparts on link prediction and node classification (Chami et al., 2019); knowledge graphs likewise benefit from multi-relational hyperbolic embeddings (Balazevic et al., 2019). In vision, hyperbolic image embeddings improve retrieval and classification under class hierarchies (Khrulkov et al., 2020), and fully hyperbolic CNNs (HCNN) generalize convolution, normalization, and MLR directly in the Lorentz model, yielding strong encoder-side gains (Bdeir et al., 2024). Hyperbolic modeling has also been adapted to genomics, where hyperbolic genome embeddings leverage evolutionary signal to surpass Euclidean baselines across diverse benchmarks (Khan et al., 2025). In this paper, we target these two application fronts, *vision* and *genomics*.

**Hyperbolic neural networks.** Designing *fully hyperbolic* neural networks means replacing every Euclidean building block with operations that are intrinsic to a negatively curved manifold, without shuttling features back and forth between Euclidean and hyperbolic spaces. Early work on the Poincaré ball showed how to lift common layers and primitives (linear/FC maps, activations, concatenation, MLR) using gyrovector-space calculus and Möbius operations (Ganea et al., 2018; Shimizu

et al., 2020), while concurrent efforts formulated end-to-end hyperbolic models beyond simple projection heads (Chen et al., 2022). Due to the Poincaré ball's greater susceptibility to numerical instability, subsequent work tried to generalize network components to the Lorentz model (Gulcehre et al., 2019; Chami et al., 2019; Fan et al., 2023). Beyond classification heads, attention mechanisms (Gulcehre et al., 2019), graph convolutional layers (Chami et al., 2019), and fully hyperbolic generative models (Qu & Zou, 2022) were also developed in this setting. A major advance came with HCNN, which introduced Lorentz-native formulations of convolution, batch normalization, and multinomial logistic regression, bringing vision encoders fully into the hyperbolic domain (Chen et al., 2022; Bdeir et al., 2024; Yang et al., 2024b). But it still remains partially intrinsic, mixing Euclidean operations with hyperbolic ones. Our work follows this *intrinsic-first* principle and departs in key aspects: a point-to-hyperplane fully connected layer in the Lorentz model with closed-form $h$-distances for logits and GyroLBN, a Lorentz batch normalization that couples gyro-centering with gyro-scaling.

**Normalization in HNNs.** Normalization layers are central to stable and efficient training, yet extending them to curved spaces remains challenging. A general Riemannian batch normalization (RBN) based on the Fréchet mean was first made practical by differentiable solvers, enabling manifold-aware centering and variance control; however, its iterative nature can be slow in practice (Lou et al., 2020). Recently, Bdeir et al. (2024) introduced *Lorentz Batch Normalization (LBN)*, which uses the closed-form Lorentzian centroid (Law et al., 2019) for re-centering and a principled tangent-space rescaling. Concurrently, *Gyrogroup Batch Normalization (GyroBN)* generalizes RBN to manifolds with gyro-structure, performing gyro-centering and variance-controlled gyro-scaling that better preserve distributional shape under gyro-operations (Chen et al., 2025a). While LBN provides efficient Lorentzian re-centering and global rescaling, it does not explicitly align batch statistics under gyro-operations; GyroBN performs such alignment but is computationally heavier because it relies on Fréchet means. Motivated by these trade-offs, we propose *GyroLBN*, which aligns batch statistics under gyro-operations within the Lorentz model while retaining the speed and practicality of closed-form re-centering.

## 3 BACKGROUND

There are five isometric hyperbolic models that one can work with (Cannon et al., 1997). We adopt the Lorentz (hyperboloid) model due to numerical stability (Mishne et al., 2023).

**Lorentz model.** The Lorentz model embeds hyperbolic space as the upper sheet of a two-sheeted hyperboloid in $(n{+}1)$-dimensional Minkowski space. It is defined as

$$\mathbb{L}_K^n \;=\; \left\{ x \in \mathbb{R}^{n+1} \;\middle|\; \langle x, x \rangle_L = \tfrac{1}{K}, \; x_0 > 0 \right\}.$$

where $K < 0$ denotes the constant sectional curvature, and $\langle x, y \rangle_L := -x_0 y_0 + \sum_{i=1}^n x_i y_i$ is the Lorentz inner product. Following Ratcliffe (2006), we write $x = (x_0, x_s)$ with *time* coordinate $x_0$ and *space* coordinates $x_s \in \mathbb{R}^n$. The origin is $\overline{\mathbf{0}} = ((-K)^{-1/2}, 0, \dots, 0)$. The closed-form squared distance is given by $d_{\mathcal{L}}^2(\boldsymbol{x}, \boldsymbol{y}) = \|\boldsymbol{x} - \boldsymbol{y}\|_{\mathcal{L}}^2 = \frac{2}{K} - 2\langle \boldsymbol{x}, \boldsymbol{y} \rangle_{\mathcal{L}}$. Other Riemannian operators are presented in Appendix B, such as exponential and logarithmic maps, and parallel transport.

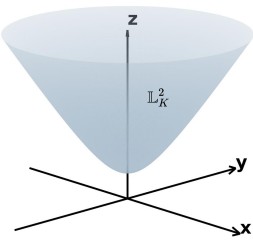

Figure 1: 2-dimensional Lorentz model in the 3-dimensional Minkowski space.

**Gyrovector space.** It forms the algebraic foundation for the hyperbolic space, as the vector space for the Euclidean space (Ungar, 2022). A gyrovector space has gyroaddition and scalar gyromultiplication, corresponding to the vector addition and scalar multiplication in Euclidean space. Recently, Chen et al. (2025c) proposed the gyroaddition and gyromultiplication over the Lorentz model:

$$x \oplus y = \mathrm{Exp}_x \left( \mathrm{PT}_{\overline{\mathbf{0}} \to x} \left( \mathrm{Log}_{\overline{\mathbf{0}}}(y) \right) \right), \quad \forall x, y \in \mathbb{L}_K^n, \tag{1}$$

$$t \odot x = \mathrm{Exp}_{\overline{\mathbf{0}}} \left( t \, \mathrm{Log}_{\overline{\mathbf{0}}}(x) \right), \quad \forall t \in \mathbb{R}, \forall x \in \mathbb{L}_K^n. \tag{2}$$

Particularly, the inverse is $\ominus x := (-1) \odot x = [x_t, -x_s]$, which satisfies $x \oplus (\ominus x) = (\ominus x) \oplus x = \overline{\mathbf{0}}$. Eqs. 1 and 2 admit closed-form expressions, which are more efficient.

## 4 INTRINSIC LORENTZ NEURAL NETWORK

We introduce the Intrinsic Lorentz Neural Network (ILNN), whose components are entirely defined by the Lorentz geometry. Specifically, we propose (i) a point-to-hyperplane fully connected (FC) layer, (ii) GyroLBN, a Lorentz batch normalization layer, and (iii) several other intrinsic modules, including log-radius concatenation, Lorentz activations, and Lorentz dropout.

### 4.1 POINT-TO-HYPERPLANE LORENTZ FULLY-CONNECTED LAYER

FC layers perform an affine transformation defined by $\boldsymbol{y} = \boldsymbol{A}\boldsymbol{x} - \boldsymbol{b}$, which can be element-wisely expressed as $y_k = \boldsymbol{a}_k \boldsymbol{x} - b_k$, where $\boldsymbol{x}, \boldsymbol{a}_k \in \mathbb{R}^n$ and $b_k \in \mathbb{R}$. Geometrically, this transformation can be interpreted as mapping the input vector $\boldsymbol{x}$ to an output score $y_k$, representing either the coordinate value or the signed distance relative to a hyperplane passing through the origin and orthogonal to the $k$-th coordinate axis in the output space $\mathbb{R}^m$ (Shimizu et al., 2020; Chen et al., 2025b). Motivated by this geometric interpretation, in this section, we first derive the Lorentz multinomial logistic regression (Lorentz MLR) to obtain the signed distance, and subsequently present the formulation of the point-to-hyperplane Lorentz fully connected (PLFC) layer.

**Lorentz MLR.** In Euclidean space, the *multinomial logistic regression* (MLR) for class $c \in \{1, \dots, C\}$ can be formulated as the logits of the Euclidean MLR classifier using the distance from instances to hyperplanes describing the class region (Lebanon & Lafferty, 2004), which can be written as

$$p(y = c \mid \boldsymbol{x}) \propto \exp(v_c(\boldsymbol{x})), \quad v_c(\boldsymbol{x}) = \text{sign}(\langle \boldsymbol{a}_c, \boldsymbol{x} - \boldsymbol{p}_c \rangle) ||\boldsymbol{a}_c|| d(\boldsymbol{x}, H_{\boldsymbol{a}_c, \boldsymbol{p}_c}), \quad \boldsymbol{a}_c \in \mathbb{R}^n, \quad (3)$$

where $H_{\boldsymbol{a}_c, \boldsymbol{p}_c} = \{\boldsymbol{x} \in \mathbb{R} : \langle \boldsymbol{a}_c, \boldsymbol{x} - \boldsymbol{p}_c \rangle = 0\}$ is the Euclidean hyperplane of class $c$ and $d(\cdot, \cdot)$ denotes Euclidean distance. Once we have the Lorentz hyperplane and closed-form point-hyperplane distance, we can extend Euclidean MLR to the Lorentz model.

In Lorentz model $\mathbb{L}_K^n$, following Bdeir et al. (2024), for $\boldsymbol{p} \in \mathbb{L}_K^n$ and $\boldsymbol{w} \in \mathcal{T}_{\boldsymbol{p}}\mathbb{L}_K^n$, the hyperplane passing through $\boldsymbol{p}$ and perpendicular to $\boldsymbol{w}$ is given by $H_{\boldsymbol{w},\boldsymbol{p}} = \{\boldsymbol{x} \in \mathbb{L}_K^n \mid \langle \boldsymbol{w}, \boldsymbol{x} \rangle_{\mathcal{L}} = 0\}$, where $\boldsymbol{w}$ should satisfy the condition $\langle \boldsymbol{w}, \boldsymbol{w} \rangle > 0$. To eliminate this condition, $\boldsymbol{w} \in \mathcal{T}_{\boldsymbol{p}}\mathbb{L}_K^n$ is parameterized by a vector $\overline{\boldsymbol{z}} \in \mathcal{T}_{\overline{\mathbf{0}}}\mathbb{L}_K^n = [0, a\boldsymbol{z}/||\boldsymbol{z}||]$, where $a \in \mathbb{R}$ and $\boldsymbol{z} \in \mathbb{R}^n$. As $\boldsymbol{w} \in \mathcal{T}_{\boldsymbol{p}}\mathbb{L}_K^n$, $\overline{\boldsymbol{z}}$ is parallel transported to $\boldsymbol{p}$, the Lorentz hyperplane defined by

$$\tilde{H}_{\boldsymbol{z},a} = \{\boldsymbol{x} \in \mathbb{L}_K^n \mid \cosh\left(\sqrt{-K}a\right)\langle \boldsymbol{z}, \boldsymbol{x}_s \rangle - \sinh\left(\sqrt{-K}a\right)||\boldsymbol{z}||\, x_t = 0\}, \quad (4)$$

where a and z represent the distance and orientation to the origin, respectively. Due to the $\boldsymbol{z} \in \mathcal{T}_{\overline{\mathbf{0}}}\mathbb{L}_K^n$ and the sign-preserving property of parallel transported, this construction naturally satisfies $\langle \boldsymbol{w}, \boldsymbol{w} \rangle > 0$. Then for $x \in \mathbb{L}_K^n$, the distance to a Lorentz hyperplane is given by

$$d_{\mathcal{L}}(\boldsymbol{x}, \tilde{H}_{\boldsymbol{z},a}) = \frac{1}{\sqrt{-K}}\left|\sinh^{-1}\left(\sqrt{-K}\,\frac{\cosh(\sqrt{-K}a)\langle \boldsymbol{z}, \boldsymbol{x}_s \rangle - \sinh(\sqrt{-K}a)||\boldsymbol{z}||_2\, x_t}{\sqrt{||\cosh(\sqrt{-K}a)\boldsymbol{z}||_2^2 - (\sinh(\sqrt{-K}a)||\boldsymbol{z}||_2)^2}}\right)\right|, \quad (5)$$

Substituting Eq. 4 and Eq. 5 into Eq. 3, we obtain that the Lorentz MLR's output logit corresponding to class c is given by

$$v_{\boldsymbol{z}_c,a_c}(\boldsymbol{x}) = \frac{1}{\sqrt{-K}}\,\text{sign}(\alpha_c)\beta_c\left|\sinh^{-1}\left(\sqrt{-K}\frac{\alpha_c}{\beta_c}\right)\right|, \quad (6)$$

$$\alpha_c = \cosh\left(\sqrt{-K}a\right)\langle \boldsymbol{z}, \boldsymbol{x}_s \rangle - \sinh\left(\sqrt{-K}a\right),$$

$$\beta_c = \sqrt{||\cosh\left(\sqrt{-K}a\right)\boldsymbol{z}||^2 - (\sinh\left(\sqrt{-K}a\right)||\boldsymbol{z}||)^2}.$$

**Lorentz fully-connected layer (PLFC).**    Shimizu et al. (2020); Chen et al. (2025b) interpreted the Euclidean FC layer as an operation that transforms the input $\boldsymbol{x}$ via $v_k(\boldsymbol{x})$, treating the output $\boldsymbol{y_k}$ as the signed distance from the hyperplane $H_{\boldsymbol{e}^{(k)},0}$ passing through the origin and orthogonal to the $k$-th axis of the output space $\mathbb{R}^m$, which can be written as

$$d^{\pm}\big(\boldsymbol{y}_k, H_{\boldsymbol{e}^{(k)},0}\big) = v_k(\boldsymbol{x}), \qquad k = 1, \ldots, m. \tag{7}$$

where $d^{\pm}(\cdot, \cdot)$ denote the signed distance in Euclidean space. We now collect the above ingredients into an *intrinsic* FC layer that maps an input $\boldsymbol{x} \in \mathbb{L}_K^n$ to an output $\boldsymbol{y} = (y_0, \boldsymbol{y}_s) \in \mathbb{L}_K^m$. Let $\{(\boldsymbol{z}_k, a_k)\}_{k=1}^m$ be learnable hyperplane parameters and define $v_k(\boldsymbol{x}) = v_{\boldsymbol{z}_k, a_k}(\boldsymbol{x})$ as in Eq. 6. Matching each $v_k(\boldsymbol{x})$ to the signed $h$-distance from $\boldsymbol{y}$ to the $k$-th *coordinate hyperplane* $\bar{H}_{e^{(k)},0} = \{\boldsymbol{x} = (x_0, x_1, \ldots, x_m)^T \in \mathbb{L}_K^m \mid \langle \boldsymbol{e}^{(k)}, x \rangle = \boldsymbol{x}_k = 0, k = 1, 2 \ldots, m\}$ fixes the spatial coordinates, while the time coordinate follows from the hyperboloid constraint.

**Theorem 1** (PLFC layer). *Let $\boldsymbol{x} \in \mathbb{L}_K^n$, $Z = \{\boldsymbol{z}_k\}_{k=1}^m \subset \mathbb{R}^n$ and $a = \{a_k\}_{k=1}^m \subset \mathbb{R}$. The* point–to–hyperplane Lorentz fully connected layer $\mathrm{PLFC}_K : \mathbb{L}_K^n \to \mathbb{L}_K^m$ *is*

$$v_k(\boldsymbol{x}) = v_{\boldsymbol{z}_k, a_k}(\boldsymbol{x}), \quad k = 1, \ldots, m, \tag{8a}$$

$$y_{s,k} = \frac{1}{\sqrt{-K}} \sinh\big(\sqrt{-K}\, v_k(\boldsymbol{x})\big), \tag{8b}$$

$$y = \left[ \sqrt{(-K)^{-1} + \|\boldsymbol{y}_s\|_2^2}, \boldsymbol{y}_s \right], \tag{8c}$$

*where $\boldsymbol{y}_s = (y_{s,1}, \ldots, y_{s,m})^{\top}$. In the flat-space limit $K \to 0$, equation 8 reduces to the Euclidean affine map $\boldsymbol{y} = Av + b$ with $A_k = \boldsymbol{z}_k^{\top}$ and bias $b = a$.*

It can be shown that the signed distance from $\boldsymbol{y}$ to each Lorentz hyperplane passing through the origin and orthogonal to the $k$-th coordinate axis is given by $v_k(\boldsymbol{x})$, as detailed in the Appendix E.1, thereby fulfilling the properties described above.

**Gyro-bias.**    A learnable offset $\boldsymbol{b} \in \mathbb{L}_K^m$ can be added intrinsically via the gyroaddition, $\boldsymbol{y} \leftarrow \boldsymbol{y} \oplus \boldsymbol{b}$, yielding the final PLFC output.

**Discussion.**    The previous Lorentz fully connected layer (LFC) used in HCNN was formed by

$$\mathbf{y} = \left[ \sqrt{|\phi(W\mathbf{x}, \mathbf{v})|^2 - 1/K}, \phi(W\mathbf{x}, \mathbf{v}) \right],$$

where $\mathbf{x} \in \mathcal{L}_K^n$ and $W\mathbf{x}$ is a standard matrix–vector product in $\mathbb{R}^n$. This $W\mathbf{x}$ is defined using the ambient linear structure of the Minkowski space, not any operation on the Lorentz manifold so itself only partially intrinsic. The PLFC depends *only* on Lorentzian operations, avoids tangent-space linearisation, and enjoys closed-form gradients, making it an efficient and curvature-consistent replacement for Euclidean FC layers in hyperbolic networks. More detailed discussions are in Appendix D and a theoretical advantage of PLFC over LFC is shown in Appendix E.2.

## 4.2    Gyrogroup Lorentz Batch Normalization (GyroLBN)

Batch Normalization (BN) facilitates training by normalizing batch statistics. Recently, Bdeir et al. (2024) proposed LBN, which uses a Lorentzian centroid to efficiently compute the batch mean. However, their approach fails to normalize sample statistics. Besides, Chen et al. (2025a;c) extended BN into manifolds based on gyrogroup structures, referred to GyroBN (Chen et al., 2025a). Although it can normalize sample mean/variance on Lorentz spaces, the involved Fréchet mean could be inefficient. Therefore, we propose *GyroLBN* to combine the gyrogroup normalization with Lorentzian centroid & radius statistics, retaining GyroBN's effectiveness while eliminating computationally expensive Fréchet mean.

**Batch centering and dispersion.**    Generally, the Fréchet mean must be solved iteratively, which significantly delays the training speed. Therefore, following Law et al. (2019); Bdeir et al. (2024), with $\boldsymbol{x}_i \in \mathbb{L}_K^n$, $\nu_i \geq 0, \sum_{i=1}^m \nu_i > 0$ and a batch of features $B = \{x_i \in \mathbb{L}_K^n\}_{i=1}^m$, is given by, we define the batch mean by Lorentzian centroid which can be calculated efficiently in closed form (Law et al., 2019):

$$\mu_B = \frac{\sum_{i=1}^m \nu_i \boldsymbol{x}_i}{\sqrt{-K}\left| \|\sum_{i=1}^m \nu_i \boldsymbol{x}_i\|_{\mathcal{L}} \right|}, \tag{9}$$

which solves $\min_{\mu_B \in \mathbb{L}_K^n} \sum_{i=1}^m \nu_i d_{\mathcal{L}}^2(\boldsymbol{x}_i, \mu_B)$. Moreover, the mean is not weighted, which gives $\nu_i = \frac{1}{m}$.

As for dispersion, we adopt the Fréchet variance $\sigma^2 \in \mathbb{R}^+$, defined as the expected squared Lorentzian distance between a point $\boldsymbol{x}_i$ and the mean $\mu_B$, and given by $\sigma_B^2 = \frac{1}{m} \sum_{i=1}^m d_{\mathcal{L}}^2(\boldsymbol{x}_i, \mu)$ (Kobler et al., 2022).

**Normalization map.** Denote learned scale $\gamma \in \mathbb{R}_{>0}$ (per-channel) and learned bias $\beta \in L_K^d$; For each sample $x$, the GyroLBN output is

$$\forall i \leq N, \quad \tilde{x}_i \leftarrow \overbrace{\beta \oplus}^{\text{Biasing}} \left( \overbrace{\frac{\gamma}{\sqrt{\sigma_B + \epsilon}} \odot}^{\text{Scaling}} \left( \overbrace{\ominus \mu_B \oplus x_i}^{\text{Centering}} \right) \right), \qquad \varepsilon > 0. \tag{10}$$

where $\oplus$ denotes left gyroaddition Eq. 1, $\ominus$ denotes gyro inverse, and $\odot$ denotes gyro scalar product Eq. 2. Unless otherwise stated, during inference, we maintain per-channel running statistics by updating the Lorentzian centroid (mean) and dispersion (variance) with momentum and, at test time, substitute these for batch stats.

**Discussion.** GyroLBN unifies the *gyrogroup* normalization paradigm of GyroBN with the *efficient* Lorentz statistics of LBN, differing from GyroBN only in the choice of statistics by replacing Fréchet-based batch estimates with closed-form Lorentzian centroid and variance while retaining the same gyrogroup centering and scaling scheme. Compared to Fréchet variance used in generic Riemannian BN (Lou et al., 2020; Chen et al., 2022), the mean-radius statistic avoids iterative solvers and proved numerically stable in our vision/genomics settings. It therefore (i) remains fully intrinsic, (ii) avoids iterative Fréchet solvers (important for large batches and 2D convolutions), and (iii) integrates naturally with gyro-additive residual/bias layers used in our encoder. Compared to LBN and Fréchet-based GyroBN (Lou et al., 2020), GyroLBN consistently reduced wall-clock time while improving accuracy in our settings, as shown in Table 4.

## 4.3 OTHER LORENTZ MODULES

**Log-radius concatenation.** When concatenating $N$ Lorentz patches, each with $(1 + d)$ coordinates (one time and $d$ spatial), naively stacking the $Nd$ spatial components biases the resulting feature norm toward higher dimensions: the expected radius grows with $Nd$, skewing subsequent layers. We introduce a *log-radius*–preserving concatenation that makes the expected *log* spatial radius invariant to the number of concatenated blocks. Let $v \in \mathbb{R}^{n_i}$ denote the spatial part of a block and assume its radius factorizes as $\|v\| = \sigma\sqrt{T}$ with $T \sim \chi_{n_i}^2$. Then

$$\mathbb{E}[\log \|v\|] = \log \sigma + \tfrac{1}{2}\left(\psi\left(\tfrac{n_i}{2}\right) + \log 2\right),$$

where $\psi$ is the digamma function. To keep $\mathbb{E}[\log \|v\|]$ constant across dimensions, we scale each block's spatial part by

$$s(n, n_i) = \exp\left(\frac{1}{2}\left[\psi\left(\frac{n}{2}\right) - \psi\left(\frac{n_i}{2}\right)\right]\right),$$

with $n = Nd$ the total post-concat spatial dimension and $n_i = d$ the per-block spatial dimension. Concretely, given per-window tensors $(t_1, \ldots, t_N)$ (times) and $(u_1, \ldots, u_N)$ with $u_i \in \mathbb{R}^d$ (spaces), we form the scaled space $\tilde{u}_i = s\, u_i$ and recompute a single time coordinate that keeps the output on the hyperboloid:

$$t' = \sqrt{-\tfrac{1}{K} + s^2 \sum_{i=1}^{N}(t_i^2 + \tfrac{1}{K})},$$

where $k > 0$ sets the origin time via $t_0 = \sqrt{k}$ (in practice, with $K = -1$, $k = 1$). The final concatenated vector is $[t', \tilde{u}_1, \ldots, \tilde{u}_N] \in \mathbb{R}^{1+Nd}$. This *log-radius* alignment (i) is parameter-free, (ii) is robust to heavy-tailed radii because it matches *geometric* means, (iii) preserves the Lorentz constraint by design, and (iv) avoids domination by any single wide block, improving stability as kernel size or channel count grows.

**Lorentz convolutional layer.** We use channel-last feature map representations throughout HCNNs, and add the Lorentz model's time component as an additional channel dimension, following Bdeir et al. (2024). A hyperbolic feature map can be defined as an ordered set of n-dimensional hyperbolic vectors, where every spatial position contains a vector that can be combined with its neighbors

The convolutional layer can be formulated as a matrix multiplication between a linearized kernel and the concatenation of values within its receptive field (Shimizu et al., 2020). Then, we extend this definition by replacing the Euclidean FC and concatenation with our PLFC and *log-radius*–preserving concatenation.

Let $\mathbf{x} = \{\boldsymbol{x}_{h,w} \in \mathbb{L}_K^n\}_{h,w=1}^{H,W}$ be an input hyperbolic feature map and let $\tilde{H} \times \tilde{W}$ denotes the kernel size with stride $\delta$. For each spatial location $(h, w)$ we gather the patch $\{\boldsymbol{x}_{h+\delta\tilde{h},\, w+\delta\tilde{w}}\}_{\tilde{h},\tilde{w}=1}^{\tilde{H},\tilde{W}} \subset \mathbb{L}_K^n$, pad with the origin $(\sqrt{1/(-K)}, 0, \ldots, 0)$ and concatenate these $\tilde{H}\tilde{W}$ vectors by the *log-radius* scheme introduced above. We defined the Lorentz convolutional layer as

$$\boldsymbol{y}_{h,w} = \text{PLFC}(\text{LogCat}(\{\boldsymbol{x}_{h'+\delta\tilde{h},w'+\delta\tilde{w}} \in \mathbb{L}_K^n\}_{\tilde{h},\tilde{w}=1}^{\tilde{H},\tilde{W}})), \tag{11}$$

where $h'$ and $w'$ denote the starting position, and LogCat denotes our *log-radius*–preserving concatenation.

**Lorentz dropout.** To regularize features without leaving the manifold, we adopt a dropout operator that acts on Lorentz coordinates and then *reprojects* to the hyperboloid. Concretely, during training, we apply an elementwise Bernoulli mask to the current representation $x \in \mathbb{L}_K^n$ (probability $p$ of zeroing each entry), yielding $\tilde{x}$ in ambient $\mathbb{R}^{n+1}$. Because naive masking can violate the hyperboloid constraint and the positivity of the time component, we immediately map back via a projection $\text{proj}_{\mathcal{L}}(\tilde{x})$ that restores $\langle x, x \rangle_L = 1/K$ and $x_0 > 0$. At evaluation, the operator is the identity. In practice, we observe that the "mask & project" scheme outperforms the variant "log–exp" scheme that applies a log map to $T_0\mathbb{L}_K^n$, performs Euclidean dropout, and then returns via the exponential map, because the nonlinearity of $\exp_0$ couples all coordinates so that masking any tangent component perturbs the entire point after mapping back to the manifold. It is parameter-free, numerically stable, and compatible with subsequent intrinsic layers (e.g., GyroLBN and PLFC), since its output again lies on $\mathbb{L}_K^n$.

**Lorentz activation.** Following Bdeir et al. (2024), we define activations *directly* in the Lorentz model by acting on the spatial coordinates only and then recomputing the time coordinate from the hyperboloid constraint. For example, given $x = (x_0, x_s) \in L_K^n$ and an elementwise nonlinearity activation function ReLU, *Lorentz ReLU* can be defined as

$$\text{L}-\text{ReLU}(x) \;=\; \left[ \sqrt{\tfrac{1}{-K} + \|\text{ReLU}(x_s)\|_2^2} \,,\; \text{ReLU}(x_s) \right].$$

## 5 EXPERIMENT

We evaluate ILNN on image classification dataset CIFAR-10/100 (Krizhevsky et al., 2009) and genomics (TEB (Khan et al., 2025), GUE (Zhou et al., 2023)) and compare against Euclidean and multiple hyperbolic baselines under matched training recipes. For fairness, each Euclidean backbone is translated to the hyperbolic setting via *one-to-one* module replacement, keeping depth/width, parameter count, and schedule as close as possible. All models are implemented in PyTorch (Paszke et al., 2019) with 32-bit precision. We fix curvature to $K = -1$ and train with Riemannian optimizers from Geoopt (Kochurov et al., 2020): RiemannianSGD for CIFAR and RiemannianAdam for genomics. Unless otherwise stated, results are averaged over five random seeds and reported as mean $\pm$ std; classification uses accuracy on CIFAR-10/100, and MCC on genomics. All experiments are conducted on NVIDIA A100 80GB GPUs. Additional configuration details (batch size, learning rate) appear in Appendix F.

### 5.1 IMAGE CLASSIFICATION

**Experimental setup.** Following the evaluation setup of HCNN (Bdeir et al., 2024) and symmetric space (Nguyen et al., 2025), we benchmark ILNN on CIFAR-10 and CIFAR-100. For each method, we instantiate a ResNet-18 backbone (He et al., 2015) in the corresponding geometry: The original ResNet-18 with BN and linear classifier (Euclidean); Euclidean encoder + hyperbolic head (Poincaré or Lorentz); all layers fully to the target manifold (HCNN (Bdeir et al., 2024) and our ILNN). The relative distortion from input manifold to classifier (Bdeir et al., 2024) is comparable across runs ($\delta_{\text{rel}}$=0.26 for CIFAR-10; 0.23 for CIFAR-100).

**Main results.** Table 1 summarizes test accuracy as mean$\pm$sd over five runs. ILNN attains the highest average accuracy on both datasets, 95.36% on CIFAR-10 and 78.41% on CIFAR-100, exceeding the Euclidean ResNet-18 baseline by +0.22 and +0.69 percentage points (pp), respectively. Relative to the strongest prior hyperbolic competitor (HCNN-Lorentz), ILNN also improves by +0.22 pp on CIFAR-10 and +0.34 pp on CIFAR-100. Although ILNN exhibits a slightly larger standard deviation than

Table 1: Classification accuracy (%) of ResNet-18 models. We estimate the mean and standard deviation from five runs. The best performance is highlighted in bold (higher is better).

| | CIFAR-10 ($\delta_{\text{rel}} = 0.26$) | CIFAR-100 ($\delta_{\text{rel}} = 0.23$) |
|---|---|---|
| Euclidean (He et al., 2015) | $95.14 \pm 0.12$ | $77.72 \pm 0.15$ |
| Hybrid Poincaré (Guo et al., 2022) | $95.04 \pm 0.13$ | $77.19 \pm 0.50$ |
| Poincaré ResNet (Van Spengler et al., 2023) | $94.51 \pm 0.15$ | $76.60 \pm 0.32$ |
| Euclidean-Poincaré-H (Fan et al., 2023) | $81.72 \pm 7.84$ | $44.35 \pm 2.93$ |
| Euclidean-Poincaré-G (Ganea et al., 2018) | $95.14 \pm 0.11$ | $77.78 \pm 0.09$ |
| Euclidean-Poincaré-B (Nguyen et al., 2025) | $95.23 \pm 0.08$ | $77.78 \pm 0.15$ |
| Hybrid Lorentz (Bdeir et al., 2024) | $94.98 \pm 0.12$ | $78.03 \pm 0.21$ |
| HCNN Lorentz (Bdeir et al., 2024) | $95.14 \pm 0.08$ | $78.07 \pm 0.17$ |
| ILNN (ours) | $\mathbf{95.36 \pm 0.13}$ | $\mathbf{78.41 \pm 0.23}$ |

HCNN-Lorentz, on CIFAR-10, even the conservative lower bound of ILNN (95.23%) exceeds the upper bound of HCNN (95.22%); on CIFAR-100, the lower bound of ILNN is specifically 78.24%, equal to the upper bound of HCNN. These gains demonstrate that our intrinsic approach preserves the geometry of the Lorentz model without distortion: by combining point-to-hyperplane FC and GyroLBN within fully negative-curvature space, ILNN leverages the native manifold structure more effectively than others.

**Visualization.** We visualize embeddings by mapping network outputs from the Lorentz model to the Poincaré ball and, in parallel, applying the logarithmic map at the origin to view them in the tangent plane; colors indicate output label prediction. In Figure 2 (CIFAR-10), the ILNN embeddings form ten compact clusters, whose areas are visibly smaller than the corresponding clusters produced by HCNN. Only marginal colour bleeding occurs at the boundaries, indicating that the decision margins learned by the point-to-hyperplane FC induce margins aligned with the data geometry. The effect becomes even more pronounced on the harder CIFAR-100 task, as shown in Figure 3. Despite packing 100 classes into the same 2D manifold, ILNN still yields dense colour "islands" with clear gaps in between, whereas HCNN exhibits overlapping clouds and several mixed-colour zones. These visual trends are consistent with the quantitative improvements in Table 1.

## 5.2 GENOMIC CLASSIFICATION

**Experimental setup.** We evaluate on the most challenging subsets of the **TEB** and **GUE** genomics benchmarks, **specifically those on which the published HCNN (Khan et al., 2025) does not surpass a Euclidean convolutional baseline.** For every dataset, we instantiate four models that share the same convolutional stem and classifier width: In our comparisons, the Euclidean CNN uses standard convolutions, Euclidean BN, and a linear classifier; HCNN-S employs Lorentz modules with a single global curvature $K$ across the network; HCNN-M uses Lorentz modules with layer-wise curvatures $K$; and ILNN is fully intrinsic, featuring GyroLBN and a point-to-hyperplane FC with gyro-bias, while fixing a global curvature $K = -1$. All hyperbolic models are optimized with RiemannianAdam (lr $10^{-3}$); the Euclidean baseline uses Adam with an identical schedule. We measure performance by the Matthews correlation coefficient (MCC).

**Main results.** Table 2 shows that ILNN achieves the best score on every task. On the two TEB pseudogene sets, it improves over the Euclidean baseline by +9.6 and +13.0 pp, and still exceeds the stronger HCNN-S by +2.0 and +8.8 pp, respectively, demonstrating clear gains where previous hyperbolic models were already competitive. The advantage is even more striking on GUE. For Covid-variant classification, HCNN collapses (MCC 36.7/14.8) much lower than the Euclidean model, whereas ILNN matches and slightly surpasses the Euclidean score (64.8 vs. 63.6). On promoter-related tasks, ILNN consistently raises the bar: Tata core-promoter detection jumps from 79.9 (best prior) to 83.9, and the most difficult "all" split goes from 67.2 to 70.9. Across the board, ILNN tightens the worst-case gap between hyperbolic and Euclidean models and converts several cases of HCNN under-performance into clear wins. These results confirm that fully intrinsic design choices, point-to-hyperplane FC, GyroLBN, and other proposed components, translate into tangible accuracy gains on real-world genomic data.

Table 2: Model performance (MCC) on all real-world genomics datasets, averaged over five random seeds (mean $\pm$ standard deviation). The two highest-scoring models are in bold. * denotes that the result was reproduced under the same setting.

| Benchmark | Task | Dataset | Model | | | |
|---|---|---|---|---|---|---|
| | | | Euclidean CNN | Hyperbolic HCNN-S | Hyperbolic HCNN-M | Hyperbolic ILNN |
| TEB | Pseudogenes | processed | $60.66_{\pm0.82}$ | $\underline{68.30_{\pm0.93}}$ | $65.41_{\pm5.54}$ | $\mathbf{70.26_{\pm0.32}}$ |
| | | unprocessed | $51.94_{\pm2.69}$ | $56.13_{\pm0.56}$ | $\underline{58.36_{\pm1.80}}$ | $\mathbf{64.90_{\pm0.74}}$ |
| GUE | Covid Variant Classification | Covid | $\underline{63.62_{\pm1.34}}$* | $36.71_{\pm9.69}$ | $14.81_{\pm0.46}$ | $\mathbf{64.76_{\pm0.54}}$ |
| | Core Promoter Detection | tata | $78.26_{\pm2.85}$ | $79.54_{\pm1.61}$ | $\underline{79.87_{\pm2.50}}$ | $\mathbf{83.90_{\pm0.53}}$ |
| | | notata | $\underline{66.60_{\pm1.07}}$ | $66.52_{\pm0.28}$ | $65.95_{\pm0.51}$ | $\mathbf{72.59_{\pm0.69}}$ |
| | | all | $66.47_{\pm0.74}$ | $65.26_{\pm1.11}$ | $\underline{67.16_{\pm0.55}}$ | $\mathbf{70.89_{\pm0.43}}$ |
| | Promoter Detection | tata | $78.58_{\pm3.39}$ | $\underline{79.74_{\pm2.66}}$ | $78.77_{\pm0.78}$ | $\mathbf{83.26_{\pm1.90}}$ |
| | | notata | $\underline{90.81_{\pm0.51}}$ | $89.86_{\pm0.76}$ | $90.28_{\pm0.37}$ | $\mathbf{92.48_{\pm0.35}}$ |
| | | all | $\underline{88.00_{\pm0.39}}$ | $87.60_{\pm0.51}$ | $87.93_{\pm0.76}$ | $\mathbf{91.34_{\pm0.38}}$ |

## 5.3 GRAPHS

**Experimental setup.** To further evaluate the effectiveness of our method, we extend it to three widely-used graph datasets (AIRPORT, CORA and PUBMED), which exhibit intricate topological and hierarchical relationships, making them an ideal testbed for evaluating the effectiveness of hyperbolic networks (e.g., HGNN, HGCN, HGAT, HAN, HNN++ and Hypformer). We choose the Hypformer as baseline and replace the Linear layer in Hypformer.

**Main results.** The quantitative results are summarized in Table 3. Overall, hyperbolic models consistently outperform their Euclidean counterparts on all three benchmarks, confirming that negatively curved representa-

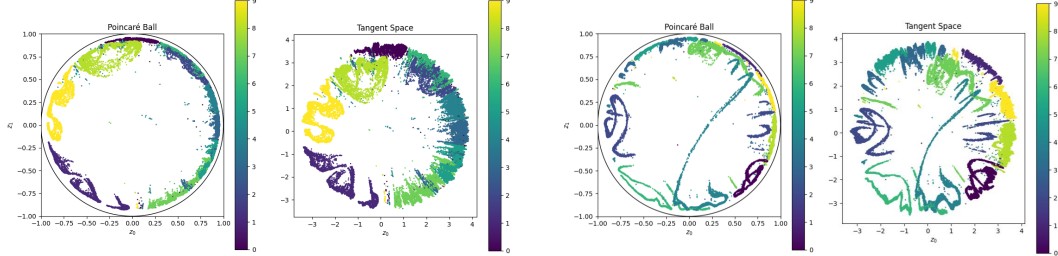

Figure 2: Embedding visualization of CIFAR-10 dataset in Poincaré and Tangent Space. Colors represent labels. HCNN (94.98, left) and ILNN (95.48, right).

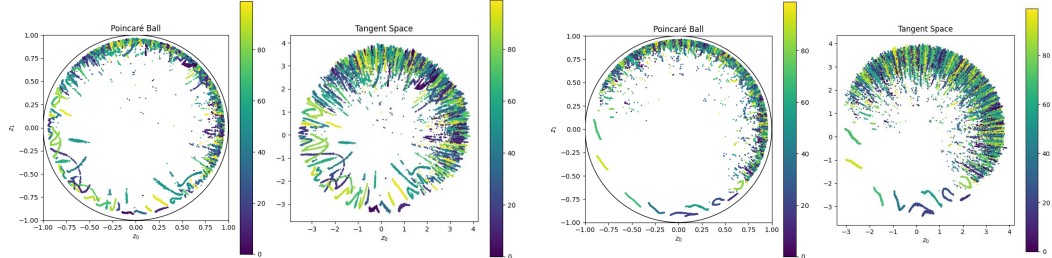

Figure 3: Embedding visualization of CIFAR-100 dataset in Poincaré and Tangent Space. Colors represent labels. HCNN (77.67, left) and ILNN (78.64, right).

tions are well-suited for graphs with rich hierarchical structure. Among existing methods, HAN, HNN++, and SGFormer constitute the strongest baselines, while Hypformer further improves their performance. Building on this backbone, Hypformer+PLFC achieves the best results on all datasets, reaching $96.03\%\pm0.34$ on AIRPORT, $85.68\%\pm0.19$ on CORA, and $82.52\%\pm0.33$ on PUBMED. Compared with the original Hypformer, this corresponds to absolute gains of $+1.03$, $+0.68$, and $+1.22$ percentage points, respectively. The standard deviations remain small, and the confidence intervals of Hypformer+PLFC are largely separated from those of competing methods on AIRPORT and PUBMED, indicating that the observed gains are statistically stable. These results demonstrate that replacing the Euclidean linear classifier in Hypformer with our hyperbolic point-to-hyperplane PLFC layer provides a simple yet effective way to enhance graph representation learning on graph benchmarks.

## 5.4 ABLATION STUDY

To better understand the contributions of each architectural component, we conduct an ablation study on our two main innovations: the point-to-hyperplane fully connected head (PLFC) and the Gyrogroup Lorentz Batch Normalization (GyroLBN). We compare them against their Lorentz counterparts (LFC, LBN) as well as GyroBN, measuring both classification accuracy across CIFAR-10 and genomics tasks, and training efficiency to disentangle the effects of the classifier design and the normalization strategy, and to assess whether their combination leads to complementary improvements.

Table 3: Testing results (Accuracy) on AIRPORT, CORA and PUBMED. The best results are in bold, respectively.

| Models | AIRPORT | CORA | PUBMED |
|---|---|---|---|
| GCN (Kipf & Welling, 2017) | $81.4 \pm 0.6$ | $81.3 \pm 0.3$ | $78.1 \pm 0.2$ |
| GAT (Veličković et al., 2017) | $81.5 \pm 0.3$ | $83.0 \pm 0.7$ | $79.0 \pm 0.3$ |
| SGC (Wu et al., 2019) | $82.1 \pm 0.5$ | $80.1 \pm 0.2$ | $78.7 \pm 0.1$ |
| HGNN (Liu et al., 2019) | $84.7 \pm 1.0$ | $77.1 \pm 0.8$ | $78.3 \pm 1.2$ |
| HGCN (Chami et al., 2019) | $89.3 \pm 1.2$ | $76.5 \pm 0.6$ | $78.0 \pm 1.0$ |
| HGAT (Chami et al., 2019) | $89.6 \pm 1.0$ | $77.4 \pm 0.7$ | $78.3 \pm 1.4$ |
| GraphFormer (Ying et al., 2021) | $88.1 \pm 1.2$ | $60.0 \pm 0.5$ | $73.3 \pm 0.7$ |
| GraphTrans (Wu et al., 2021) | $94.3 \pm 0.6$ | $77.6 \pm 0.8$ | $77.5 \pm 0.7$ |
| GraphGPS (Rampášek et al., 2022) | $94.5 \pm 0.9$ | $73.0 \pm 1.4$ | $72.8 \pm 1.4$ |
| FPS-T (Cho et al., 2023) | $96.0 \pm 0.6$ | $82.3 \pm 0.7$ | $78.5 \pm 0.6$ |
| HAN (Gulcehre et al., 2019) | $92.9 \pm 0.6$ | $83.1 \pm 0.5$ | $79.0 \pm 0.6$ |
| HNN++ (Shimizu et al., 2020) | $92.3 \pm 0.3$ | $82.8 \pm 0.6$ | $79.9 \pm 0.4$ |
| F-HNN (Chen et al., 2022) | $93.0 \pm 0.7$ | $81.0 \pm 0.7$ | $77.5 \pm 0.8$ |
| NodeFormer (Wu et al., 2022) | $80.2 \pm 0.6$ | $82.2 \pm 0.9$ | $79.9 \pm 1.0$ |
| SGFormer (Wu et al., 2023) | $92.9 \pm 0.5$ | $83.2 \pm 0.9$ | $80.0 \pm 0.8$ |
| Hypformer (Yang et al., 2024b) | $95.0 \pm 0.5$ | $85.0 \pm 0.3$ | $81.3 \pm 0.3$ |
| Hypformer+PLFC | $\mathbf{96.0 \pm 0.3}$ | $\mathbf{85.7 \pm 0.2}$ | $\mathbf{82.5 \pm 0.3}$ |

Table 4: Ablation on PLFC and GyroLBN. Fit time denotes the training time of the CIFAR-10 dataset per epoch.

| Benchmark | Task | Dataset | Model | | | |
|---|---|---|---|---|---|---|
| | | | LFC LBN | LFC GyroLBN | PLFC GyroBN | PLFC GyroLBN |
| | CIFAR-10 | | $95.14_{\pm 0.08}$ | $95.19_{\pm 0.15}$ | $95.28_{\pm 0.17}$ | $\mathbf{95.36_{\pm 0.13}}$ |
| GUE | Core Promoter Detection | tata | $78.26_{\pm 2.85}$ | $81.33_{\pm 3.19}$ | $80.89_{\pm 3.11}$ | $\mathbf{83.90_{\pm 0.53}}$ |
| | | notata | $66.60_{\pm 1.07}$ | $71.92_{\pm 0.52}$ | $72.22_{\pm 0.82}$ | $\mathbf{72.59_{\pm 0.69}}$ |
| | | all | $66.47_{\pm 0.74}$ | $69.74_{\pm 1.3}$ | $70.14_{\pm 0.45}$ | $\mathbf{70.89_{\pm 0.43}}$ |
| | Promoter Detection | tata | $78.58_{\pm 3.39}$ | $80.46_{\pm 0.99}$ | $81.16_{\pm 1.99}$ | $\mathbf{83.26_{\pm 1.90}}$ |
| | | notata | $90.81_{\pm 0.51}$ | $91.88_{\pm 1.01}$ | $91.67_{\pm 0.49}$ | $\mathbf{92.48_{\pm 0.35}}$ |
| | | all | $88.00_{\pm 0.39}$ | $90.28_{\pm 1.04}$ | $91.02_{\pm 0.56}$ | $\mathbf{91.34_{\pm 0.38}}$ |
| | Fit Time(s) | | 142 | 125 | 314 | 169 |

**PLFC vs. LFC.**   Holding the normalizer fixed, replacing the Lorentz fully connected head with the point-to-hyperplane fully connected head consistently improves accuracy. With GyroLBN, PLFC outperforms LFC on CIFAR-10 (95.36 vs. 95.19, +0.17) and on all six genomics subsets, with gains of +2.57 on *tata* core-promoter detection (83.90 vs. 81.33), +0.67 on *notata* (72.59 vs. 71.92), and +0.97 on the *all* split (70.89 vs. 69.74); for promoter detection, the improvements are +2.80 (*tata*: 83.26 vs. 80.46), +0.60 (*notata*: 92.48 vs. 91.88), and +1.06 (*all*: 91.34 vs. 90.28). Although the PLFC variant with GyroBN also performs strongly, exceeding the LFC baseline by more than +3 on all genomics subsets and matching or surpassing it on CIFAR-10 (95.28 vs. 95.14). These results indicate that decision functions based on point-to-hyperplane distance provide a more effective inductive bias than affine logits in both vision and genomics.

**GyroLBN vs. LBN, GyroBN.**   Under the same FC, GyroLBN improves over LBN across all tasks. With the LFC head, CIFAR-10 increases from 95.14 to 95.19; on genomics, the gains range from +1.07 to +5.32 (e.g., core-promoter *notata*: 71.92 vs. 66.60). With the PLFC head, GyroLBN is superior to GyroBN on every dataset, including *notata* in core-promoter detection, where it reaches 72.59 versus 72.22. The improvements span +0.32 to +3.01 on genomics and +0.08 on CIFAR-10. Training time measurements show that GyroLBN attains these gains with favorable efficiency, running faster than PLFC with GyroBN (169 s vs. 314 s) and even faster than LBN for the comparison of LFC between GyroLBN and LBN (125 s vs. 142 s). Overall, GyroLBN offers a better accuracy and efficiency trade-off than both LBN and GyroBN in our settings.

Furthermore, to ensure a fair comparison with GyroBN, we further vary the number of Fréchet mean iterations used by the normalizers (1, 2, 5, 10, and a fixed-point solve denoted by $\infty$; see Appendix G). Across all iteration cases, *PLFC+GyroLBN* remains the best-performing configuration on every genomics split, exhibiting the effectiveness of our GyroLBN again.

## 6   CONCLUSION

This study presented the Intrinsic Lorentz Neural Network, an architecture whose computations remain entirely within the Lorentz model of hyperbolic space. We introduced a point-to-hyperplane fully connected layer that converts signed hyperbolic distances into logits, together with GyroLBN and log-radius concatenation for numerically stable normalization and feature aggregation. Integrated into a coherent network, these components yield superior performance on CIFAR-10, CIFAR-100, and two challenging genomics benchmarks, surpassing Euclidean, hybrid, and prior hyperbolic baselines while preserving competitive training cost. The results underscore the value of keeping every layer intrinsic to the manifold and provide practical building blocks for future work on representation learning in negatively curved geometries.

## REPRODUCIBILITY STATEMENT

All theoretical results are established under explicit conditions. Experimental details are given in Appendix F. The code is available at this url.

## ETHICS STATEMENT

This work only uses publicly available datasets and does not involve human subjects or sensitive information. We identify no specific ethical concerns.

## ACKNOWLEDGEMENTS

This work was supported by EU Horizon project ELLIOT (No. 101214398) and by the FIS project GUIDANCE (No. FIS2023-03251). We acknowledge CINECA for awarding high-performance computing resources under the ISCRA initiative, and the EuroHPC Joint Undertaking for granting access to Leonardo at CINECA, Italy.

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

APPENDIX

## A    USE OF LARGE LANGUAGE MODELS

We use large language models to aid or polish writing.

## B    OPERATIONS IN THE LORENTZ MODEL

**Setup and notation**    Fix a negative curvature $K < 0$. Let $\langle \cdot, \cdot \rangle_{\mathcal{L}}$ denote the Minkowski bilinear form with signature $(-, +, \ldots, +)$ on $\mathbb{R}^{n+1}$, and write $\| \cdot \|_{\mathcal{L}} = \sqrt{\langle \cdot, \cdot \rangle_{\mathcal{L}}}$ for the induced (Riemannian) norm on tangent vectors. The $n$-dimensional hyperbolic space in the Lorentz (hyperboloid) model is

$$\mathbb{L}_K^n := \left\{ \mathbf{x} \in \mathbb{R}^{n+1} \ : \ \langle \mathbf{x}, \mathbf{x} \rangle_{\mathcal{L}} = \tfrac{1}{K}, \ \ x_0 > 0 \right\},$$

and we use $\overline{\mathbf{0}} := \left( \tfrac{1}{\sqrt{-K}}, \mathbf{0} \right)$ as the pole ("origin"). When no confusion can arise, $\| \cdot \|$ denotes the Euclidean norm in a tangent space.

**Distance**    For $\mathbf{x}, \mathbf{y} \in \mathbb{L}_K^n$, the geodesic distance inherited from Minkowski space is

$$D_{\mathcal{L}}(\mathbf{x}, \mathbf{y}) = \frac{1}{\sqrt{-K}} \ \cosh^{-1}\big( K \langle \mathbf{x}, \mathbf{y} \rangle_{\mathcal{L}} \big).$$

A useful identity for computations is the "Lorentzian chord" expression (squared distance)

$$d_{\mathcal{L}}^2(\mathbf{x}, \mathbf{y}) = \| \mathbf{x} - \mathbf{y} \|_{\mathcal{L}}^2 = \frac{2}{K} - 2 \langle \mathbf{x}, \mathbf{y} \rangle_{\mathcal{L}}.$$

Specializing to the pole $\overline{\mathbf{0}}$,

$$D_{\mathcal{L}}(\mathbf{x}, \overline{\mathbf{0}}) = \big\| \log_{\overline{\mathbf{0}}}^K(\mathbf{x}) \big\|, \qquad d_{\mathcal{L}}^2(\mathbf{x}, \overline{\mathbf{0}}) = \frac{2}{K} - 2 \langle \mathbf{x}, \overline{\mathbf{0}} \rangle_{\mathcal{L}} = \frac{2}{K} + \frac{2\,x_0}{\sqrt{-K}},$$

and, equivalently,

$$D_{\mathcal{L}}(\mathbf{x}, \overline{\mathbf{0}}) = \frac{1}{\sqrt{-K}} \ \cosh^{-1}\big( \sqrt{-K}\, x_0 \big).$$

**Tangent space**    The tangent space at $\mathbf{x} \in \mathbb{L}_K^n$ is the Minkowski-orthogonal complement of $\mathbf{x}$,

$$\mathcal{T}_{\mathbf{x}} \mathbb{L}_K^n = \big\{ \mathbf{v} \in \mathbb{R}^{n+1} \ : \ \langle \mathbf{v}, \mathbf{x} \rangle_{\mathcal{L}} = 0 \big\}.$$

Restricted to $\mathcal{T}_{\mathbf{x}} \mathbb{L}_K^n$, the metric is positive definite and coincides with the Riemannian metric of $\mathbb{L}_K^n$.

**Exponential and logarithmic maps**    For $\mathbf{z} \in \mathcal{T}_{\mathbf{x}} \mathbb{L}_K^n$,

$$\exp_{\mathbf{x}}^K(\mathbf{z}) = \cosh(\alpha)\, \mathbf{x} \ + \ \sinh(\alpha)\, \frac{\mathbf{z}}{\alpha}, \qquad \alpha = \sqrt{-K}\, \| \mathbf{z} \|_{\mathcal{L}}.$$

The inverse map $\log_{\mathbf{x}}^K : \mathbb{L}_K^n \to \mathcal{T}_{\mathbf{x}} \mathbb{L}_K^n$ sends $\mathbf{y} \in \mathbb{L}_K^n$ to

$$\log_{\mathbf{x}}^K(\mathbf{y}) = \frac{\cosh^{-1}(\beta)}{\sqrt{\beta^2 - 1}} \ (\mathbf{y} - \beta\, \mathbf{x}), \qquad \beta = K \langle \mathbf{x}, \mathbf{y} \rangle_{\mathcal{L}}.$$

At the pole $\overline{\mathbf{0}}$ these simplify to

$$\exp_{\overline{\mathbf{0}}}^K(\mathbf{z}) = \frac{1}{\sqrt{-K}} \left( \cosh\big( \sqrt{-K}\, \| \mathbf{z} \| \big), \ \sinh\big( \sqrt{-K}\, \| \mathbf{z} \| \big)\, \frac{\mathbf{z}}{\| \mathbf{z} \|} \right),$$

$$\log_{\overline{\mathbf{0}}}^K(\mathbf{y}) = \begin{cases} \mathbf{0}, & \mathbf{y} = \overline{\mathbf{0}}, \\[2mm] \dfrac{\cosh^{-1}(\beta)}{\sqrt{\beta^2 - 1}} (\mathbf{y} - \beta\, \overline{\mathbf{0}}), & \text{otherwise,} \end{cases} \qquad \beta := K \langle \overline{\mathbf{0}}, \mathbf{y} \rangle_{\mathcal{L}}.$$

with the convention $\mathbf{z} / \| \mathbf{z} \| = \mathbf{0}$ when $\mathbf{z} = \mathbf{0}$.

**Parallel transport**    Transporting $\mathbf{v} \in \mathcal{T}_{\mathbf{x}} \mathbb{L}_K^n$ along the geodesic from $\mathbf{x}$ to $\mathbf{y}$ yields

$$\mathrm{PT}_{\mathbf{x} \to \mathbf{y}}^K(\mathbf{v}) = \mathbf{v} - \frac{\langle \log_{\mathbf{x}}^K(\mathbf{y}), \mathbf{v} \rangle_{\mathcal{L}}}{d_{\mathcal{L}}(\mathbf{x}, \mathbf{y})} \left( \log_{\mathbf{x}}^K(\mathbf{y}) + \log_{\mathbf{y}}^K(\mathbf{x}) \right) = \mathbf{v} + \frac{\langle \mathbf{y}, \mathbf{v} \rangle_{\mathcal{L}}}{\frac{1}{-K} - \langle \mathbf{x}, \mathbf{y} \rangle_{\mathcal{L}}} (\mathbf{x} + \mathbf{y}).$$

**Lorentzian centroid and average pooling** Given points $\mathbf{x}_1, \ldots, \mathbf{x}_m \in \mathbb{L}_K^n$ with nonnegative weights $\nu_i$ (not all zero), the weighted Fréchet mean with respect to the squared Lorentzian distance, $\min_{\boldsymbol{\mu} \in \mathbb{L}_K^n} \sum_{i=1}^m \nu_i \, d_{\mathcal{L}}^2(\mathbf{x}_i, \boldsymbol{\mu})$, is obtained in closed form by

$$\boldsymbol{\mu} = \frac{\sum_{i=1}^m \nu_i \, \mathbf{x}_i}{\sqrt{-K} \left| \left\| \sum_{i=1}^m \nu_i \, \mathbf{x}_i \right\|_{\mathcal{L}} \right|} \, .$$

In neural architectures, an "average pooling" over a hyperbolic receptive field can be implemented by taking this Lorentzian centroid of the features in the field.

## C  HYPERBOLIC GYROVECTOR STRUCTURES

### C.1  GYROGROUPS

We recall the algebraic notion of a gyrogroup, which extends the concept of a group to settings where associativity is relaxed and corrected by gyrations (Ungar, 2008; 2014).

**Definition 1** (Gyrogroup). *Let $G$ be a nonempty set endowed with a binary operation $\oplus \colon G \times G \to G$. The pair $(G, \oplus)$ is a* gyrogroup *if, for all $a, b, c \in G$, the following axioms hold:*

*(G1) There exists an element $e \in G$ such that $e \oplus a = a$ (left identity).*
*(G2) For each $a \in G$ there exists $\ominus a \in G$ such that $\ominus a \oplus a = e$ (left inverse).*
*(G3) There exists a map $\mathrm{gyr}[a, b] \colon G \to G$ (the gyration generated by $a, b$) such that*

$$a \oplus (b \oplus c) = (a \oplus b) \oplus \mathrm{gyr}[a, b](c) \quad \text{(left gyroassociative law)}.$$

*(G4) $\mathrm{gyr}[a, b] = \mathrm{gyr}[a \oplus b, b]$ (left reduction law).*

*If in addition*

$$a \oplus b = \mathrm{gyr}[a, b](b \oplus a), \qquad \forall \, a, b \in G,$$

*then $(G, \oplus)$ is called* gyrocommutative.

Gyrogroups generalize groups: when all gyrations are the identity map, $(G, \oplus)$ reduces to a usual group. The lack of strict associativity is compensated by the gyration operators, which encode curvature induced nonlinearity.

### C.2  FROM GYROGROUPS TO GYROVECTOR SPACES

To model both addition and scalar multiplication in curved geometries, gyrogroups can be enriched to gyrovector spaces (Ungar, 2008; Chen et al., 2025c).

**Definition 2** (Gyrovector space). *Let $(G, \oplus)$ be a gyrocommutative gyrogroup and let $\odot \colon \mathbb{R} \times G \to G$ be a map called scalar multiplication. The triple $(G, \oplus, \odot)$ is a* gyrovector space *if, for all $a, b, c \in G$ and $s, t \in \mathbb{R}$,*

*(V1) $1 \odot a = a$.*
*(V2) $(s + t) \odot a = s \odot a \oplus t \odot a$.*
*(V3) $(st) \odot a = s \odot (t \odot a)$.*
*(V4) $\mathrm{gyr}[a, b](t \odot c) = t \odot \mathrm{gyr}[a, b](c)$.*
*(V5) $\mathrm{gyr}[s \odot a, t \odot a]$ is the identity map on $G$.*

These axioms mirror the familiar properties of vector spaces, with gyrations accounting for the deviation from linearity. In particular, (V2) and (V3) play the role of distributivity and associativity for scalar multiplication, while (V4)–(V5) guarantee a consistent interaction between gyrations and scaling.

### C.3  HYPERBOLIC GYROVECTOR SPACES

The gyro-structure over the hyperbolic space can be defined by its Riemannian operators (Ganea et al., 2018; Chen et al., 2025c). Given a hyperbolic model $\mathcal{H}_K^n$ along with $x, y, z \in \mathcal{H}_K^n$ and $t \in \mathbb{R}$, the gyroaddition and gyromultiplication are defined as

$$
\begin{aligned}
x \oplus_{\mathcal{H}} y &= \mathrm{Exp}_x \left( \mathrm{PT}_{e \to x} \left( \mathrm{Log}_e y \right) \right), \\
t \odot_{\mathcal{H}} x &= \mathrm{Exp}_e \left( t \, \mathrm{Log}_e x \right), \\
\mathrm{gyr}[x, y]z &= \ominus_{\mathcal{H}} \left( x \oplus_{\mathcal{H}} y \right) \oplus_{\mathcal{H}} \left( x \oplus_{\mathcal{H}} \left( y \oplus_{\mathcal{H}} z \right) \right),
\end{aligned}
\tag{12}
$$

where $e$ denotes the origin in $\mathcal{H}_K^n$.

On the Poincaré ball $\mathbb{P}_K^n$, such gyro-structure is known as the Möbius gyrovector space (Ungar, 2022, Ch. 6.14):

$$x \oplus_{\mathrm{M}} y = \frac{\left(1 - 2K\langle x, y\rangle - K\|y\|^2\right) x + \left(1 + K\|x\|^2\right) y}{1 - 2K\langle x, y\rangle + K^2\|x\|^2\|y\|^2},$$

$$t \odot_{\mathrm{M}} x = \frac{\tanh\left(t \tanh^{-1}(\sqrt{|K|}\|x\|)\right)}{\sqrt{|K|}} \frac{x}{\|x\|}$$

where $\ominus_{\mathrm{M}} x = -1 \odot_{\mathrm{M}} x = -x$ is the gyroinverse and $\mathbf{0}$ is the gyro identity: $\mathbf{0} \oplus_{\mathrm{M}} x = x, \forall x \in \mathbb{P}_K^n$. As shown by Chen et al. (2025c, Props. 24-25), the Lorentz gyroaddition and gyromultiplication also admit closed-form expressions:

$$x \oplus_{\mathbb{L}} y = \begin{cases} x, & y = \overline{\mathbf{0}}, \\ y, & x = \overline{\mathbf{0}}, \\ \begin{bmatrix} \frac{1}{\sqrt{|K|}} \frac{D - KN}{D + KN} \\ \frac{2\left(A_s x_s + A_y y_s\right)}{D + KN} \end{bmatrix}, & \text{Otherwise.} \end{cases} \tag{13}$$

$$t \odot_{\mathbb{L}} x = \begin{cases} \overline{\mathbf{0}}, & t = 0 \vee x = \overline{\mathbf{0}} \\ \frac{1}{\sqrt{|K|}} \begin{bmatrix} \cosh(t\theta) \\ \frac{\sinh(t\theta)}{\|x_s\|} x_s \end{bmatrix}, & \text{Otherwise,} \end{cases} \tag{14}$$

Here, $\theta = \cosh^{-1}(\sqrt{|K|}x_t)$, $A_s = ab^2 - 2Kbs_{xy} - Kan_y$ and $A_y = b(a^2 + Kn_x)$ with the following notation:

$$\begin{aligned} & a = 1 + \sqrt{|K|}x_t, b = 1 + \sqrt{|K|}y_t, \\ & n_x = \|x_s\|^2, n_y = \|y_s\|^2, s_{xy} = \langle x_s, y_s\rangle, \\ & D = a^2 b^2 - 2Kabs_{xy} + K^2 n_x n_y, \\ & N = a^2 n_y + 2abs_{xy} + b^2 n_x. \end{aligned} \tag{15}$$

In particular, the gyro identity is $\overline{\mathbf{0}}$ and the gyroinverse is $\ominus_{\mathbb{L}} x = -1 \odot_{\mathbb{L}} x = [x_t, -x_s^\top]^\top$.

## D  DISCUSSION ABOUT INTRINSIC

In our paper, "intrinsic Lorentz" refers to using only operations that are well-defined on the Lorentz model itself, rather than on its ambient Minkowski space. We define a layer ($F : \mathbb{L}_K^n \to \mathbb{L}_K^m$) intrinsic if

1. its input, output, and all intermediate states lie on some $\mathbb{L}_K^d$ (they always satisfy $\langle z, z\rangle_L = 1/K$, $z_0 > 0$).
2. it is expressed entirely in terms of the Lorentzian geometry: $\langle \cdot, \cdot\rangle_L$, the induced distance $d_L$, and operators derived from them (exp/log maps, parallel transport, gyroaddition/gyroscaling, Lorentzian centroids, etc.), without ever using arbitrary Euclidean linear maps on Lorentz vectors in the ambient space.

Under this definition, the previous Lorentz fully connected layer (LFC) used in HCNN is not intrinsic. Its update has the form

$$\mathbf{y} = \left[\sqrt{|\phi(W\mathbf{x}, \mathbf{v})|^2 - 1/K}, \phi(W\mathbf{x}, \mathbf{v})\right],$$

where $\mathbf{x} \in \mathcal{L}_K^n$, $W\mathbf{x}$ is a standard matrix–vector product in $\mathbb{R}^n$ and the operation

$$\phi(\mathbf{W}\boldsymbol{x}, \boldsymbol{v}) = \lambda\sigma(\boldsymbol{v}^T\boldsymbol{x} + \boldsymbol{b}')\frac{\mathbf{W}\psi(\boldsymbol{x}) + \boldsymbol{b}}{\|\mathbf{W}\psi(\boldsymbol{x}) + \boldsymbol{b}\|}. \tag{16}$$

This $W\mathbf{x}$ is defined using the ambient linear structure of the Minkowski space, not any operation on the Lorentz manifold itself. Because the core transformation is an ambient Minkowski multiplication rather than a Lorentzian/geodesic operation, this layer is only partially intrinsic.

By contrast, our PLFC is constructed entirely from Lorentz-geometry primitives that have closed-form definitions on $\mathcal{L}_K^n$. Each logit is the signed Lorentzian distance from the input point to a learned Lorentz hyperplane, and the output $\mathbf{y}$ is then recovered in closed form as the unique point on $\mathcal{L}_K^m$ whose signed distances to a set of coordinate hyperplanes equal these logits. All steps (hyperplane parameterization, point-to-hyperplane distance, reconstruction of $\mathbf{y}$) are expressed only via the Lorentzian inner product and distance; no ambient Euclidean affine map $W\mathbf{x} + b$ is ever applied. In this sense, PLFC is an intrinsic Lorentz FC layer.

# E PROOFS

## E.1 PROOF OF THEOREM 1

*Proof.* We work in the hyperboloid model $\mathbb{L}_K^m = \{\boldsymbol{x} = [x_t, \boldsymbol{x}_s] \in \mathbb{R}^{1+m} : \langle \boldsymbol{x}, \boldsymbol{x} \rangle_{\mathcal{L}} = 1/K, \ x_t > 0\}$ with $K < 0$ and Minkowski bilinear form $\langle [x_t, \boldsymbol{x}_s], [y_t, \boldsymbol{y}_s] \rangle_{\mathcal{L}} = -x_t y_t + \langle \boldsymbol{x}_s, \boldsymbol{y}_s \rangle$. Let $\overline{\mathbf{0}} = [(-K)^{-1/2}, \mathbf{0}]$ be the basepoint. Denote by $\mathbf{e}^{(k)} \in \mathbb{R}^m$ the $k$-th Euclidean basis vector in the spatial block and set $\boldsymbol{e}^{(k)} = [0, \mathbf{e}^{(k)}]$.

**Lorentz coordinate hyperplanes.** The $k$-th spatial axis is the geodesic through $\overline{\mathbf{0}}$ in the direction $\mathbf{e}^{(k)}$. The Lorentz *coordinate hyperplane* through $\overline{\mathbf{0}}$, orthogonal to this axis, is given by

**Definition 3.** *(Lorentz hyperplane containing $\overline{\mathbf{0}}$ and orthogonal to the k-th axis)*

$$\bar{H}_{\boldsymbol{e}^{(k)},0}^K = \big\{ \boldsymbol{x} = [x_t, \boldsymbol{x}_s]^\top \in \mathbb{L}_K^m \ \big| \ \langle \boldsymbol{e}^{(k)}, \boldsymbol{x} \rangle_{\mathcal{L}} = \langle \mathbf{e}^{(k)}, \boldsymbol{x}_s \rangle = x_{s,k} = 0 \big\}. \tag{17}$$

This is the special case of the Lorentz hyperplane $\tilde{H}_{\boldsymbol{z},a}$ in Eq. 4 with $a = 0$ and $\boldsymbol{z} = \mathbf{e}^{(k)}$ (hence $\|\boldsymbol{z}\|_2 = 1$), for which $\cosh(\sqrt{-K}a) = 1$ and $\sinh(\sqrt{-K}a) = 0$, giving $\langle \boldsymbol{z}, \boldsymbol{x}_s \rangle = x_{s,k} = 0$.

With Definition 3, the preparation for constructing $\boldsymbol{y}$ in Eq. 8 is complete.

**Derivation of $\boldsymbol{y}$.** Let $\boldsymbol{x} \in \mathbb{L}_K^n$ and $\boldsymbol{y} = [y_t, \boldsymbol{y}_s]^\top \in \mathbb{L}_K^m$ be the input and output of the PLFC layer, respectively. As in Eq. 8a, for $k = 1, \ldots, m$ we define the scores $v_k(\boldsymbol{x}) = v_{\boldsymbol{z}_k, a_k}(\boldsymbol{x})$ via the Lorentz MLR logits in Eq. 6.

To endow $\boldsymbol{y}$ with the desired property—that the *signed* hyperbolic distance from $\boldsymbol{y}$ to the $k$-th coordinate hyperplane equals $v_k(\boldsymbol{x})$—we impose the simultaneous system

$$d_{\mathcal{L}}^{\pm}\big(\boldsymbol{y}, \bar{H}_{\boldsymbol{e}^{(k)},0}^K\big) = v_k(\boldsymbol{x}), \qquad k = 1, \ldots, m. \tag{18}$$

Using Eqs. 4 and 5 with $a = 0$ and $\boldsymbol{z} = \mathbf{e}^{(k)}$ (so that the denominator equals 1), the unsigned point–to–hyperplane distance specializes to

$$d_{\mathcal{L}}\big(\boldsymbol{y}, \bar{H}_{\boldsymbol{e}^{(k)},0}^K\big) = \frac{1}{\sqrt{-K}} \big| \sinh^{-1}\big(\sqrt{-K}\, y_{s,k}\big) \big|.$$

Orienting $\bar{H}_{\boldsymbol{e}^{(k)},0}^K$ by the unit normal $\boldsymbol{e}^{(k)}$ and recalling the sign convention used in Eq. 6, the *signed* distance is therefore

$$d_{\mathcal{L}}^{\pm}\big(\boldsymbol{y}, \bar{H}_{\boldsymbol{e}^{(k)},0}^K\big) = \frac{1}{\sqrt{-K}} \sinh^{-1}\big(\sqrt{-K}\, y_{s,k}\big). \tag{19}$$

Substituting Eq. 19 into Eq. 18 yields, for each $k$,

$$\frac{1}{\sqrt{-K}} \sinh^{-1}\big(\sqrt{-K}\, y_{s,k}\big) = v_k(\boldsymbol{x}), \tag{20}$$

and hence

$$y_{s,k} = \frac{1}{\sqrt{-K}} \sinh\big(\sqrt{-K}\, v_k(\boldsymbol{x})\big), \qquad k = 1, \ldots, m, \tag{21}$$

which is exactly Eq. 8b. Collecting these coordinates gives $\boldsymbol{y}_s = (y_{s,1}, \ldots, y_{s,m})^\top$.

Finally, the time coordinate $y_t$ is fixed by the hyperboloid constraint $\langle \boldsymbol{y}, \boldsymbol{y} \rangle_{\mathcal{L}} = 1/K$, i.e.,

$$-y_t^2 + \|\boldsymbol{y}_s\|_2^2 = \frac{1}{K} \implies y_t = \sqrt{(-K)^{-1} + \|\boldsymbol{y}_s\|_2^2}, \tag{22}$$

with the positive root chosen to remain on the top sheet, which is Eq. 8c. Thus Eq. 8 follows.

**Confirmation of the existence of $\boldsymbol{y}$.** For any real scores $v_k(\boldsymbol{x})$, Eq. 21 yields real $y_{s,k}$ because sinh is entire. Since $-K > 0$, we have $(-K)^{-1} + \|\boldsymbol{y}_s\|_2^2 > 0$, so $y_t$ in Eq. 22 is real and strictly positive. Therefore $\boldsymbol{y} = [y_t, \boldsymbol{y}_s] \in \mathbb{L}_K^m$ always exists and lies on the correct sheet. Moreover, Eq. 20 guarantees that the signed distance from $\boldsymbol{y}$ to each coordinate hyperplane $\bar{H}_{\boldsymbol{e}^{(k)},0}^K$ is exactly $v_k(\boldsymbol{x})$, as required.

**Flat-space limit.** As $K \to 0^-$, we have $\sinh(\sqrt{-K}\, v) = \sqrt{-K}\, v + O(K^{3/2})$, hence $y_{s,k} \to v_k(\boldsymbol{x})$ from Eq. 21. Using Eq. 6 and the expansion $\cosh(\sqrt{-K}a) = 1 + O(K)$, $\sinh(\sqrt{-K}a) = \sqrt{-K}\, a + O(K^{3/2})$, one obtains $v_k(\boldsymbol{x}) \to \langle \boldsymbol{z}_k, \boldsymbol{x}_s \rangle - a_k$, so the spatial part reduces to the Euclidean affine map with row vectors $A_k = \boldsymbol{z}_k^\top$ and bias $b_k = a_k$, as stated below Theorem 1. $\qquad\square$

## E.2 PROOF OF THEOREM 2

**Theorem 2** (Margin preservation and contraction of PLFC and LFC). *Fix a curvature $K < 0$. For any input $x$, let the penultimate layer produce*

$$u(x) = (u_1(x), \dots, u_m(x)) \in \mathbb{R}^m,$$

*and let $c$ denote the true class. Define the pre-logit margin*

$$\Delta(x) := u_c(x) - \max_{j \neq c} u_j(x).$$

*Consider two Lorentz output-layer designs that use signed geodesic distances from the output point $y(x)$ to the coordinate hyperplanes as logits:*

- PLFC head (intrinsic). *The spatial coordinates are*

$$y_{s,k}^{\text{PLFC}}(x) = \frac{1}{\sqrt{-K}} \sinh(\sqrt{-K}\, u_k(x)),$$

  *and the signed Lorentzian distance to the $k$-th coordinate hyperplane is*

$$d_k^{\text{PLFC}}(x) = \frac{1}{\sqrt{-K}} \operatorname{asinh}(\sqrt{-K}\, y_{s,k}^{\text{PLFC}}(x)).$$

- LFC head (extrinsic / linear). *The spatial coordinates are taken directly as*

$$y_{s,k}^{\text{LFC}}(x) = u_k(x),$$

  *and the signed Lorentzian distance to the same hyperplane is*

$$d_k^{\text{LFC}}(x) = \frac{1}{\sqrt{-K}} \operatorname{asinh}(\sqrt{-K}\, u_k(x)).$$

*Define the distance-based margins*

$$\Delta^{\text{PLFC}}(x) := d_c^{\text{PLFC}}(x) - \max_{j \neq c} d_j^{\text{PLFC}}(x), \qquad \Delta^{\text{LFC}}(x) := d_c^{\text{LFC}}(x) - \max_{j \neq c} d_j^{\text{LFC}}(x).$$

*Then, for every sample $x$:*

1. *(**Margin preservation of PLFC**)*
$$\Delta^{\text{PLFC}}(x) = \Delta(x).$$

2. *(**Margin contraction of LFC**) The LFC head preserves the sign of the margin and contracts its magnitude:*
$$\operatorname{sign}(\Delta^{\text{LFC}}(x)) = \operatorname{sign}(\Delta(x)), \qquad |\Delta^{\text{LFC}}(x)| \leq |\Delta(x)|,$$

*with strict inequality $|\Delta^{\text{LFC}}(x)| < |\Delta(x)|$ whenever $\Delta(x) \neq 0$.*

*Proof.* Define

$$h(t) := \frac{1}{\sqrt{-K}} \operatorname{asinh}(\sqrt{-K}\, t), \qquad K < 0.$$

By the point–to–hyperplane distance formula in the Lorentz model, for any Lorentz point $y = (y_0, y_s)$ the signed distance to the $k$-th coordinate hyperplane is exactly $h(y_{s,k})$.

*(1) PLFC preserves the margin.* For the PLFC head we have

$$y_{s,k}^{\text{PLFC}}(x) = \frac{1}{\sqrt{-K}} \sinh(\sqrt{-K}\, u_k(x)),$$

and thus

$$d_k^{\text{PLFC}}(x) = h(y_{s,k}^{\text{PLFC}}(x)) = \frac{1}{\sqrt{-K}} \operatorname{asinh}\left(\sqrt{-K} \cdot \frac{1}{\sqrt{-K}} \sinh(\sqrt{-K}\, u_k(x))\right) = \frac{1}{\sqrt{-K}} \operatorname{asinh}\left(\sinh\left(\sqrt{-K}\, u_k(x)\right)\right).$$

Since $\sinh : \mathbb{R} \to \mathbb{R}$ is a bijection with inverse asinh, we have $\operatorname{asinh}(\sinh z) = z$ for all $z \in \mathbb{R}$, hence

$$d_k^{\text{PLFC}}(x) = u_k(x).$$

Consequently,

$$\Delta^{\text{PLFC}}(x) = d_c^{\text{PLFC}}(x) - \max_{j \neq c} d_j^{\text{PLFC}}(x) = u_c(x) - \max_{j \neq c} u_j(x) = \Delta(x),$$

which proves (1).

*(2) LFC contracts the margin.* For the LFC head we have $y_{s,k}^{\mathrm{LFC}}(x) = u_k(x)$, hence

$$d_k^{\mathrm{LFC}}(x) = h\big(u_k(x)\big).$$

We first record two basic properties of $h$. Differentiating,

$$h'(t) = \frac{\mathrm{d}}{\mathrm{d}t}\Big[\frac{1}{\sqrt{-K}}\,\mathrm{asinh}\big(\sqrt{-K}\,t\big)\Big] = \frac{1}{\sqrt{-K}}\cdot\frac{\sqrt{-K}}{\sqrt{1+(-K)t^2}} = \frac{1}{\sqrt{1+(-K)t^2}},$$

so $h'(t) > 0$ for all $t$ and $h'(t) \leq 1$ with $h'(t) = 1$ if and only if $t = 0$. Therefore, $h$ is strictly increasing and 1-Lipschitz.

Let $j^\star$ be any index of a maximizer of the competing pre-logits:

$$j^\star \in \arg\max_{j\neq c} u_j(x).$$

Since $h$ is strictly increasing, the same index maximizes the distance-based logits:

$$\max_{j\neq c} d_j^{\mathrm{LFC}}(x) = \max_{j\neq c} h(u_j(x)) = h\big(u_{j^\star}(x)\big).$$

Hence

$$\Delta(x) = u_c(x) - u_{j^\star}(x), \qquad \Delta^{\mathrm{LFC}}(x) = h\big(u_c(x)\big) - h\big(u_{j^\star}(x)\big).$$

If $\Delta(x) = 0$, then $u_c(x) = u_{j^\star}(x)$ and consequently $\Delta^{\mathrm{LFC}}(x) = 0$, so the conclusion holds trivially. Suppose now $\Delta(x) \neq 0$ and, without loss of generality, set

$$a := u_c(x), \qquad b := u_{j^\star}(x), \qquad a \neq b.$$

Assume $a > b$; the case $a < b$ is analogous by symmetry. Using the fundamental theorem of calculus,

$$h(a) - h(b) = \int_b^a h'(t)\,\mathrm{d}t.$$

Because $h'(t) > 0$ for all $t$, we have $h(a) - h(b) > 0$, so $\mathrm{sign}(h(a) - h(b)) = \mathrm{sign}(a - b)$. Moreover, since $h'(t) \leq 1$ everywhere and $h'(t) < 1$ for all $t \neq 0$, the integrand is strictly less than 1 on a subset of $[b, a]$ with positive measure whenever $a \neq b$. Thus

$$0 < h(a) - h(b) = \int_b^a h'(t)\,\mathrm{d}t < \int_b^a 1\,\mathrm{d}t = a - b.$$

Taking absolute values yields

$$\big|h(a) - h(b)\big| < |a - b|.$$

Substituting back $a = u_c(x)$ and $b = u_{j^\star}(x)$, we obtain

$$\mathrm{sign}\big(\Delta^{\mathrm{LFC}}(x)\big) = \mathrm{sign}\big(\Delta(x)\big), \qquad \big|\Delta^{\mathrm{LFC}}(x)\big| = \big|h(a) - h(b)\big| < |a - b| = \big|\Delta(x)\big|.$$

Combining with the $\Delta(x) = 0$ case, this gives

$$\big|\Delta^{\mathrm{LFC}}(x)\big| \leq \big|\Delta(x)\big|,$$

with strict inequality whenever $\Delta(x) \neq 0$, which proves (2). $\qquad\square$

## F  IMPLEMENTATION DETAILS

### F.1  DATASETS

All these datasets exhibit hierarchical class relations and high hyperbolicity (low $\delta_{rel}$) as shown in Table 5, making the use of hyperbolic models well-motivated.

**Image Datasets**  For image classification, we adopt the standard benchmarks CIFAR-10 and CIFAR-100(Krizhevsky et al., 2009). CIFAR-10 and CIFAR-100 each contain 60,000 $32 \times 32$ color images drawn from 10 and 100 classes, respectively. Following the PyTorch setup and (Bdeir et al., 2024), we use 50,000 images for training and 10,000 for testing. CIFAR-10 and CIFAR-100 are standard proxies for visual object recognition whose categories naturally admit semantic hierarchies (e.g., animal $\rightarrow$ mammal $\rightarrow$ dog $\rightarrow$ specific breed). In CIFAR-100, this is made explicit by grouping the 100 fine-grained classes into 20 coarse superclasses in the original dataset design. This hierarchical structure has been extensively verified in prior hyperbolic vision work Bdeir et al. (2024); Nguyen et al. (2025) on CIFAR-10/100

**Gene Datasets** For genomic sequence classification, we evaluate on the Transposable Elements Benchmark (TEB) (Khan et al., 2025) and the Genome Understanding Evaluation (GUE) (Zhou et al., 2023) suite. TEB is a curated, multi-species collection of binary classification tasks spanning seven transposable-element families across retrotransposons, DNA transposons, and pseudogenes; we follow the authors' released preprocessing and data partitions. GUE aggregates 28 datasets covering seven biologically meaningful tasks—including transcription-factor binding, epigenetic mark prediction, promoter and splice-site detection—with sequences ranging roughly from 70 to 1000 base pairs originating from yeast, mouse, human, and viral genomes. Unless otherwise noted, we adopt the official train/validation/test splits and report the Matthews correlation coefficient (MCC) as our primary metric. The TEB and GUE suites are constructed directly from natural genomic sequences and inherit the biological hierarchies of their domains. TEB tasks span multiple transposable-element families across retrotransposons, DNA transposons, and pseudogenes, which themselves sit in a multi-level taxonomic hierarchy (orders → superfamilies → families). GUE aggregates datasets for transcription-factor binding, promoter and core-promoter detection, splice-site prediction, and COVID-variant classification across several species. These tasks are all manifestations of hierarchical regulatory structure (e.g., motifs → modules → promoters → gene expression).

Table 5: Hyperbolicity values of the datasets used in our experiments ($\delta_{\mathrm{rel}}$).

| Benchmark | Task | Dataset | $\delta_{\mathbf{rel}}$ |
|---|---|---|---|
| CIFAR | Image classification | CIFAR-10 | 0.26 |
| | | CIFAR-100 | 0.23 |
| TEB | Pseudogenes | processed | 0.19 |
| | Pseudogenes | unprocessed | 0.16 |
| GUE | Covid variant classification | covid | 0.42 |
| | Core Promoter detection | all | 0.23 |
| | | notata | 0.21 |
| | | tata | 0.14 |
| | Promoter detection | all | 0.26 |
| | | notata | 0.26 |
| | | tata | 0.14 |

## F.2 SETTINGS

Table 6 summarizes the hyperparameters used to train the model. We additionally note a dataset-specific choice regarding normalization statistics. On *CIFAR-10/100* we enable running statistics: we maintain per-channel running statistics by updating the Lorentzian centroid (mean) and dispersion (variance) with a momentum term and, at test time, substitute these running estimates for the batch statistics. In contrast, on the *TEB* and *GUE* genomic suites, we disable running statistics entirely, because enabling them consistently led to a collapse of MCC after a few dozen epochs. For these genomic datasets, we therefore compute statistics on-the-fly from each evaluation batch (i.e., no moving averages are used at test time). Compared with natural images, the genomic tasks exhibit stronger distributional non-stationarity (heterogeneous sequence lengths and tasks) and higher batch-to-batch variability. Under these conditions, momentum-based running estimates accumulate bias and lag behind the true data distribution; in a Lorentzian normalization layer, a biased centroid and underestimated dispersion can over- or under-normalize timelike features, shrinking margins and destabilizing optimization.

## G MORE ABLATION STUDIES

Table 7 above expands the ablation to the number of Fréchet mean iterations used by each normalizer while keeping all other components fixed. Across all GUE datasets and both tasks, *PLFC+GyroLBN* achieves the best accuracy under every iteration budget, and the relative ordering of methods is unchanged as the budget increases. Moving from 1 to 2 and 5 iterations yields modest but consistent gains, whereas 10 steps and the fixed-point solution (cells with gray background, denoted by $\infty$) offer only marginal improvements at a higher computational cost. These trends indicate that the advantage of GyroLBN stems from the normalization rule itself rather than from merely computing a tighter Fréchet mean. In practice, allocating a small budget of two to five iterations recovers nearly all of the attainable accuracy while preserving efficiency, making *PLFC+GyroLBN* the most effective and economical choice in our setting.

Table 6: Summary of hyperparameters used in different datasets.

| Hyperparameter | CIFAR-10&100 | TEB | GUE |
|---|---|---|---|
| Epochs | 200 | 150 | 150 |
| Batch size | 128 | 256 | 256 |
| Learning rate (LR) | 1e-1 | 8e-4 | 9e-4 |
| Drop LR epochs | 60, 120, 160 | 100,130 | 100,130 |
| Drop LR gamma | 0.2 | 0.1 | 0.1 |
| Weight decay | 5e-4 | 6e-3 | 5e-3 |
| Optimizer | (Riemannian)SGD | (Riemannian)Adam | (Riemannian)Adam |
| Floating point precision | 32 bit | 32 bit | 32 bit |
| GPU type | A100 80G | A100 80G | A100 80G |
| Num. GPUs | 1 | 1 | 1 |
| Hyperbolic curvature $K$ | $-1$ | $-1$ | $-1$ |
| Dropout rate | 0.05 | 0.05 | 0.05 |

Table 7: Ablation study on the number of Fréchet mean iterations. The symbol $\infty$ indicates that iterations are performed until convergence, which is the setting used in our main ablation experiments.

| Benchmark | Task | Dataset | Iteration | LFC LBN | LFC GyroLBN | PLFC GyroBN | PLFC GyroLBN |
|---|---|---|---|---|---|---|---|
| GUE | Core Promoter Detection | tata | 1 | | | $80.09_{\pm1.90}$ | |
| | | | 2 | | | $81.38_{\pm2.74}$ | |
| | | | 5 | $78.26_{\pm2.85}$ | $81.33_{\pm3.19}$ | $79.47_{\pm2.21}$ | $\mathbf{83.90_{\pm0.53}}$ |
| | | | 10 | | | $81.27_{\pm1.75}$ | |
| | | | $\infty$ | | | $80.89_{\pm3.11}$ | |
| | | notata | 1 | | | $71.99_{\pm0.68}$ | |
| | | | 2 | | | $71.06_{\pm0.49}$ | |
| | | | 5 | $66.60_{\pm1.07}$ | $71.92_{\pm0.52}$ | $71.65_{\pm0.79}$ | $\mathbf{72.59_{\pm0.69}}$ |
| | | | 10 | | | $71.12_{\pm1.75}$ | |
| | | | $\infty$ | | | $72.22_{\pm1.44}$ | |
| | | all | 1 | | | $70.47_{\pm0.85}$ | |
| | | | 2 | | | $70.42_{\pm0.49}$ | |
| | | | 5 | $66.47_{\pm0.74}$ | $69.74_{\pm1.3}$ | $70.75_{\pm0.45}$ | $\mathbf{70.89_{\pm0.43}}$ |
| | | | 10 | | | $69.81_{\pm0.67}$ | |
| | | | $\infty$ | | | $70.14_{\pm0.45}$ | |
| | Promoter Detection | tata | 1 | | | $81.71_{\pm1.80}$ | |
| | | | 2 | | | $80.19_{\pm1.90}$ | |
| | | | 5 | $78.58_{\pm3.39}$ | $80.46_{\pm0.99}$ | $81.69_{\pm3.90}$ | $\mathbf{83.26_{\pm1.90}}$ |
| | | | 10 | | | $79.79_{\pm1.91}$ | |
| | | | $\infty$ | | | $81.16_{\pm1.99}$ | |
| | | notata | 1 | | | $92.15_{\pm0.76}$ | |
| | | | 2 | | | $91.89_{\pm0.71}$ | |
| | | | 5 | $90.81_{\pm0.51}$ | $91.88_{\pm1.01}$ | $92.19_{\pm0.41}$ | $\mathbf{92.48_{\pm0.35}}$ |
| | | | 10 | | | $92.07_{\pm0.63}$ | |
| | | | $\infty$ | | | $91.67_{\pm0.56}$ | |
| | | all | 1 | | | $90.45_{\pm0.72}$ | |
| | | | 2 | | | $91.01_{\pm0.99}$ | |
| | | | 5 | $88.00_{\pm0.39}$ | $90.28_{\pm1.04}$ | $90.80_{\pm0.73}$ | $\mathbf{91.34_{\pm0.38}}$ |
| | | | 10 | | | $91.18_{\pm0.83}$ | |
| | | | $\infty$ | | | $91.02_{\pm0.56}$ | |

