# INTRINSIC LORENTZ NEURAL NETWORK

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

| GUE | Core Promoter Detection | tata notata all | $78.26_{\pm2.85}$ $66.60_{\pm1.07}$ $66.47_{\pm0.74}$ | $81.33_{\pm3.19}$ $71.92_{\pm0.52}$ $69.74_{\pm1.3}$ | $80.89_{\pm3.11}$ $72.22_{\pm0.82}$ $70.14_{\pm0.45}$ | $\mathbf{83.90}_{\pm\mathbf{0.53}}$ $\mathbf{72.59}_{\pm\mathbf{0.69}}$ $\mathbf{70.89}_{\pm\mathbf{0.43}}$ |
| | Promoter Detection | tata notata all | $78.58_{\pm3.39}$ $90.81_{\pm0.51}$ $88.00_{\pm0.39}$ | $80.46_{\pm0.99}$ $91.88_{\pm1.01}$ $90.28_{\pm1.04}$ | $81.16_{\pm1.99}$ $91.67_{\pm0.49}$ $91.02_{\pm0.56}$ | $\mathbf{83.26}_{\pm\mathbf{1.90}}$ $\mathbf{92.48}_{\pm\mathbf{0.35}}$ $\mathbf{91.34}_{\pm\mathbf{0.38}}$ |
| | Fit Time(s) | | 142 | 125 | 314 | 169 |

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

**Lorentz transformations** A matrix $\mathbf{A} \in \mathbb{R}^{(n+1) \times (n+1)}$ is a Lorentz transformation if it preserves the Minkowski product: $\langle \mathbf{A}\mathbf{x}, \mathbf{A}\mathbf{y} \rangle_{\mathcal{L}} = \langle \mathbf{x}, \mathbf{y} \rangle_{\mathcal{L}}$ for all $\mathbf{x}, \mathbf{y}$. Such matrices form the Lorentz group $\mathbf{O}(1, n)$ (equivalently, $\mathbf{A}^\top \eta \, \mathbf{A} = \eta$ for the Minkowski metric $\eta$). Restricting to transformations that map the upper sheet to itself yields the time-orientation–preserving subgroup

$$\mathbf{O}^+(1, n) = \left\{ \mathbf{A} \in \mathbf{O}(1, n) \ : \ (\mathbf{A}\mathbf{x})_0 > 0 \text{ for all } \mathbf{x} \in \mathbb{L}_K^n \right\},$$

which acts by isometries on $\mathbb{L}_K^n$.

Every $\mathbf{A} \in \mathbf{O}^+(1, n)$ admits a polar decomposition into a Lorentz rotation and a Lorentz boost, $\mathbf{A} = \mathbf{R}\mathbf{B}$. The rotation fixes the time axis and rotates spatial coordinates:

$$\mathbf{R} = \begin{bmatrix} 1 & \mathbf{0}^\top \\ \mathbf{0} & \tilde{\mathbf{R}} \end{bmatrix}, \qquad \tilde{\mathbf{R}} \in \mathbf{SO}(n).$$

A boost with velocity vector $\mathbf{v} \in \mathbb{R}^n$ ($\|\mathbf{v}\| < 1$) has the block form

$$\mathbf{B} = \begin{bmatrix} \gamma & -\gamma \, \mathbf{v}^\top \\ -\gamma \, \mathbf{v} & \mathbf{I}_n + \dfrac{\gamma^2}{1 + \gamma} \, \mathbf{v}\mathbf{v}^\top \end{bmatrix}, \qquad \gamma = \frac{1}{\sqrt{1 - \|\mathbf{v}\|^2}}.$$

(Equivalently, the spatial block can be written $\mathbf{I}_n + \frac{\gamma - 1}{\|\mathbf{v}\|^2} \mathbf{v}\mathbf{v}^\top$ when $\mathbf{v} \neq \mathbf{0}$.)

## C  GYROVECTOR STRUCTURES ON THE LORENTZ MODEL

### C.1  GYROGROUPS

We recall the algebraic notion of a gyrogroup, which extends the concept of a group to settings where associativity is relaxed and corrected by gyrations (Ungar, 2008; 2014).

**Definition 1** (Gyrogroup). *Let $G$ be a nonempty set endowed with a binary operation $\oplus \colon G \times G \to G$. The pair $(G, \oplus)$ is a* gyrogroup *if, for all $a, b, c \in G$, the following axioms hold:*

*(G1) There exists an element $e \in G$ such that $e \oplus a = a$ (left identity).*
*(G2) For each $a \in G$ there exists $\ominus a \in G$ such that $\ominus a \oplus a = e$ (left inverse).*
*(G3) There exists a map $\mathrm{gyr}[a, b] \colon G \to G$ (the gyration generated by $a, b$) such that*

$$a \oplus (b \oplus c) = (a \oplus b) \oplus \mathrm{gyr}[a, b](c) \quad \text{(left gyroassociative law).}$$

*(G4) $\mathrm{gyr}[a, b] = \mathrm{gyr}[a \oplus b, b]$ (left reduction law).*

*If in addition*

$$a \oplus b = \mathrm{gyr}[a, b](b \oplus a), \qquad \forall \, a, b \in G,$$

*then $(G, \oplus)$ is called* gyrocommutative.

Gyrogroups generalize groups: when all gyrations are the identity map, $(G, \oplus)$ reduces to a usual group. The lack of strict associativity is compensated by the gyration operators, which encode curvature induced nonlinearity.

### C.2  FROM GYROGROUPS TO GYROVECTOR SPACES

To model both addition and scalar multiplication in curved geometries, gyrogroups can be enriched to gyrovector spaces (Ungar, 2008; Nguyen et al., 2022).

**Definition 2** (Gyrovector space). *Let $(G, \oplus)$ be a gyrocommutative gyrogroup and let $\odot \colon \mathbb{R} \times G \to G$ be a map called scalar multiplication. The triple $(G, \oplus, \odot)$ is a* gyrovector space *if, for all $a, b, c \in G$ and $s, t \in \mathbb{R}$,*

*(V1) $1 \odot a = a$, $0 \odot a = e$, $t \odot e = e$, and $(-1) \odot a = \ominus a$.*

*(V2)* $(s + t) \odot a = s \odot a \oplus t \odot a$.

*(V3)* $(st) \odot a = s \odot (t \odot a)$.

*(V4)* $\mathrm{gyr}[a, b](t \odot c) = t \odot \mathrm{gyr}[a, b](c)$.

*(V5)* $\mathrm{gyr}[s \odot a, t \odot a]$ *is the identity map on* $G$.

These axioms mirror the familiar properties of vector spaces, with gyrations accounting for the deviation from linearity. In particular, (V2) and (V3) play the role of distributivity and associativity for scalar multiplication, while (V4)–(V5) guarantee a consistent interaction between gyrations and scaling.

### C.3 GYROSTRUCTURES INDUCED BY RIEMANNIAN GEOMETRY

The above algebraic objects can be constructed on a Riemannian manifold from the exponential and logarithmic maps at a distinguished origin. Following Nguyen et al. (2022); Nguyen & Yang (2023); Chen et al. (2025a), let $(\mathcal{M}, g)$ be a complete Riemannian manifold with identity element $E \in \mathcal{M}$. Denote by $\mathrm{Exp}_x$ and $\mathrm{Log}_x$ the Riemannian exponential and logarithmic maps at $x \in \mathcal{M}$, and by $\mathrm{PT}_{x \to y}$ the parallel transport from $T_x\mathcal{M}$ to $T_y\mathcal{M}$. For $P, Q, R \in \mathcal{M}$ and $t \in \mathbb{R}$, define

$$P \oplus Q = \mathrm{Exp}_P\big(\mathrm{PT}_{E \to P}(\mathrm{Log}_E Q)\big), \tag{12}$$

$$t \odot P = \mathrm{Exp}_E\big(t \, \mathrm{Log}_E P\big), \tag{13}$$

$$\ominus P = (-1) \odot P = \mathrm{Exp}_E\big(-\mathrm{Log}_E P\big), \tag{14}$$

$$\mathrm{gyr}[P, Q]R = (\ominus(P \oplus Q)) \oplus \big(P \oplus (Q \oplus R)\big). \tag{15}$$

The induced gyro inner product, norm, and distance are

$$\langle P, Q \rangle_{\mathrm{gr}} = \big\langle \mathrm{Log}_E P, \mathrm{Log}_E Q \big\rangle_{T_E\mathcal{M}}, \tag{16}$$

$$\|P\|_{\mathrm{gr}} = \sqrt{\langle P, P \rangle_{\mathrm{gr}}}, \tag{17}$$

$$d_{\mathrm{gr}}(P, Q) = \big\|\ominus P \oplus Q\big\|_{\mathrm{gr}}. \tag{18}$$

Under mild regularity assumptions, $(\mathcal{M}, \oplus, \odot)$ forms a gyrovector space and the gyrodistance $d_{\mathrm{gr}}$ coincides with the geodesic distance on a wide class of manifolds, including constant curvature spaces (Nguyen et al., 2022; Chen et al., 2025a). In Euclidean space, these constructions reduce exactly to standard vector addition, scalar multiplication, and the Euclidean metric.

### C.4 CONSTANT CURVATURE MODEL AND MÖBIUS OPERATIONS

For hyperbolic geometry it is convenient to introduce the constant curvature model

$$\mathcal{M}_K = \begin{cases} \mathbb{P}_K^n, & K < 0, \\ \mathbb{R}^n, & K = 0, \end{cases}$$

where $\mathbb{P}_K^n$ is the Poincaré ball of curvature $K < 0$ and radius $1/\sqrt{-K}$. On $\mathbb{P}_K^n$ the gyrostructures in Eqs. 12 and 13 admit closed form expressions known as Möbius addition and Möbius scalar multiplication (Ungar, 2008; Ganea et al., 2018; Skopek et al., 2020).

Let $x, y \in \mathbb{P}_K^n$ and set $c = -K > 0$. The Möbius addition is

$$x \oplus_K y = \frac{\big(1 + 2c\langle x, y \rangle + c\|y\|^2\big)x + \big(1 - c\|x\|^2\big)y}{1 + 2c\langle x, y \rangle + c^2\|x\|^2\|y\|^2}, \tag{19}$$

and the Möbius scalar multiplication is

$$t \otimes_K x = \begin{cases} \dfrac{1}{\sqrt{c}} \, \tanh\big(t \, \mathrm{artanh}(\sqrt{c}\,\|x\|)\big) \, \dfrac{x}{\|x\|}, & x \neq 0, \\ 0, & x = 0. \end{cases} \tag{20}$$

Equipped with $\oplus_K$ and $\otimes_K$, the ball $\mathbb{P}_K^n$ is a real gyrovector space whose gyrodistance coincides with the hyperbolic geodesic distance.

### C.5 INDUCED GYROVECTOR OPERATIONS ON THE LORENTZ MODEL

The Lorentz model $\mathbb{L}_K^n$ is isometric to the Poincaré ball $\mathbb{P}_K^n$ via a standard mapping. Let $K < 0$ and denote $r = 1/\sqrt{-K}$. For $x = (x_0, x_s) \in \mathbb{L}_K^n$ with $x_0 > 0$ and $-x_0^2 + \|x_s\|^2 = 1/K$, define

$$\psi \colon \mathbb{L}_K^n \to \mathbb{P}_K^n, \qquad \psi(x) = \frac{x_s}{x_0 + r}, \tag{21}$$

$$\psi^{-1} \colon \mathbb{P}_K^n \to \mathbb{L}_K^n, \qquad \psi^{-1}(u) = \left(\frac{1 + c\|u\|^2}{1 - c\|u\|^2} \, r, \ \frac{2u}{1 - c\|u\|^2}\right), \tag{22}$$

where $c = -K > 0$. The map $\psi$ is a Riemannian isometry between $(\mathbb{L}_K^n, g_K)$ and $(\mathbb{P}_K^n, g_K)$ (Ratcliffe, 2006; Skopek et al., 2020).

We transfer the gyrovector structure from the ball to the hyperboloid by conjugation with $\psi$. For $x, y \in \mathbb{L}_K^n$ and $t \in \mathbb{R}$, define

$$x \oplus_K^{\mathcal{L}} y = \psi^{-1}\big(\psi(x) \oplus_K \psi(y)\big), \tag{23}$$

$$t \otimes_K^{\mathcal{L}} x = \psi^{-1}\big(t \otimes_K \psi(x)\big). \tag{24}$$

Substituting the explicit expressions Eqs. 19 and 20 and the coordinate maps Eqs. 21 and 22 yields closed form formulas for $x \oplus_K^{\mathcal{L}} y$ and $t \otimes_K^{\mathcal{L}} x$ in Lorentz coordinates. By construction, $(\mathbb{L}_K^n, \oplus_K^{\mathcal{L}}, \otimes_K^{\mathcal{L}})$ is a gyrovector space that is isomorphic to $(\mathbb{P}_K^n, \oplus_K, \otimes_K)$. In particular, the gyrodistance induced by Eq. 23 agrees with the hyperbolic geodesic distance on $\mathbb{L}_K^n$, and the gyroaddition and gyro scalar multiplication act as hyperbolic analogues of Euclidean vector addition and Euclidean scaling.

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

^{\mathrm{PLFC}}(x) = \frac{1}{\sqrt{-K}} \operatorname{asinh}\big(\sqrt{-K}\, y_{s,k}^{\mathrm{PLFC}}(x)\big).$$

- **LFC head (extrinsic / linear).** *The spatial coordinates are taken directly as*

$$y_{s,k}^{\mathrm{LFC}}(x) = u_k(x),$$

*and the signed Lorentzian distance to the same hyperplane is*

$$d_k^{\mathrm{LFC}}(x) = \frac{1}{\sqrt{-K}}\,\mathrm{asinh}\big(\sqrt{-K}\,u_k(x)\big).$$

*Define the distance-based margins*

$$\Delta^{\mathrm{PLFC}}(x) := d_c^{\mathrm{PLFC}}(x) - \max_{j \neq c} d_j^{\mathrm{PLFC}}(x), \qquad \Delta^{\mathrm{LFC}}(x) := d_c^{\mathrm{LFC}}(x) - \max_{j \neq c} d_j^{\mathrm{LFC}}(x).$$

*Then, for every sample $x$:*

1. *(**Margin preservation of PLFC**)*

$$\Delta^{\mathrm{PLFC}}(x) = \Delta(x).$$

2. *(**Margin contraction of LFC**) The LFC head preserves the sign of the margin and contracts its magnitude:*

$$\mathrm{sign}\big(\Delta^{\mathrm{LFC}}(x)\big) = \mathrm{sign}\big(\Delta(x)\big), \qquad \big|\Delta^{\mathrm{LFC}}(x)\big| \leq \big|\Delta(x)\big|,$$

*with strict inequality $\big|\Delta^{\mathrm{LFC}}(x)\big| < \big|\Delta(x)\big|$ whenever $\Delta(x) \neq 0$.*

*Proof.* Define

$$h(t) := \frac{1}{\sqrt{-K}}\,\mathrm{asinh}\big(\sqrt{-K}\,t\big), \qquad K < 0.$$

By the point–to–hyperplane distance formula in the Lorentz model, for any Lorentz point $y = (y_0, y_s)$ the signed distance to the $k$-th coordinate hyperplane is exactly $h(y_{s,k})$.

*(1) PLFC preserves the margin.* For the PLFC head we have

$$y_{s,k}^{\mathrm{PLFC}}(x) = \frac{1}{\sqrt{-K}}\,\sinh\big(\sqrt{-K}\,u_k(x)\big),$$

and thus

$$d_k^{\mathrm{PLFC}}(x) = h\big(y_{s,k}^{\mathrm{PLFC}}(x)\big) = \frac{1}{\sqrt{-K}}\,\mathrm{asinh}\Big(\sqrt{-K}\cdot\frac{1}{\sqrt{-K}}\,\sinh\big(\sqrt{-K}\,u_k(x)\big)\Big) = \frac{1}{\sqrt{-K}}\,\mathrm{asinh}\big(\sinh\big(\sqrt{-K}\,u_k(x)\big)\big).$$

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

| | | | 2 | | | 81.38$\pm$2.74 | |
| | | | 5 | | | 79.47$\pm$2.21 | |
| | | | 10 | | | 81.27$\pm$1.75 | |
| | | | $\infty$ | | | 80.89$\pm$3.11 | |
| | | notata | 1 | 66.60$\pm$1.07 | 71.92$\pm$0.52 | 71.99$\pm$0.68 | **72.59$\pm$0.69** |
| | | | 2 | | | 71.06$\pm$0.49 | |
| | | | 5 | | | 71.65$\pm$0.79 | |
| | | | 10 | | | 71.12$\pm$1.75 | |
| | | | $\infty$ | | | 72.22$\pm$1.44 | |
| | | all | 1 | 66.47$\pm$0.74 | 69.74$\pm$1.3 | 70.47$\pm$0.85 | **70.89$\pm$0.43** |
| | | | 2 | | | 70.42$\pm$0.49 | |
| | | | 5 | | | 70.75$\pm$0.45 | |
| | | | 10 | | | 69.81$\pm$0.67 | |
| | | | $\infty$ | | | 70.14$\pm$0.45 | |
| | Promoter Detection | tata | 1 | 78.58$\pm$3.39 | 80.46$\pm$0.99 | 81.71$\pm$1.80 | **83.26$\pm$1.90** |
| | | | 2 | | | 80.19$\pm$1.90 | |
| | | | 5 | | | 81.69$\pm$3.90 | |
| | | | 10 | | | 79.79$\pm$1.91 | |
| | | | $\infty$ | | | 81.16$\pm$1.99 | |
| | | notata | 1 | 90.81$\pm$0.51 | 91.88$\pm$1.01 | 92.15$\pm$0.76 | **92.48$\pm$0.35** |
| | | | 2 | | | 91.89$\pm$0.71 | |
| | | | 5 | | | 92.19$\pm$0.41 | |
| | | | 10 | | | 92.07$\pm$0.63 | |
| | | | $\infty$ | | | 91.67$\pm$0.56 | |
| | | all | 1 | 88.00$\pm$0.39 | 90.28$\pm$1.04 | 90.45$\pm$0.72 | **91.34$\pm$0.38** |
| | | | 2 | | | 91.01$\pm$0.99 | |
| | | | 5 | | | 90.80$\pm$0.73 | |
| | | | 10 | | | 91.18$\pm$0.83 | |
| | | | $\infty$ | | | 91.02$\pm$0.56 | |