# OpenReview forum: "Intrinsic Lorentz Neural Network"
_ICLR.cc/2026/Conference — ICLR 2026 Poster_

### Official Review · Reviewer_UjqT · 2025-10-26

**Soundness:** 2
**Presentation:** 3
**Contribution:** 2
**Rating:** 6
**Confidence:** 3

**Summary:**

This paper introduces the Intrinsic Lorentz Neural Network (ILNN), which is a hyperbolic NN architecture that keeps all computations inside the Lorentz space instead of mixing Euclidean operations with manifold operations. They propose point-to-hyperplane Lorentz fully connected (PLFC) that computes the space-dimension of the output as signed distance to hyperplanes, and GyroLBN that re-centers and scales the output with efficient statistics. They show improved in CIFAR datasets and genomics datasets, with ablations that show effectiveness of PLFC and GyroLBN against LBN, GyroBN, and LFC.

**Strengths:**

1. The paper demonstrates substantial improvements over HCNN and Euclidean CNN in genomics tasks
2. The visualization in Figure 2/3 shows better separated clusters than HCNN
3. Using intrinsic hyperbolic operations instead of relying implicitly on Euclidean operations is an interesting direction that could result in more principled model designs

**Weaknesses:**

1. The authors claimed that LFC "applies Euclidean mappings to spatial components while treating the time-like coordinate separately" on line 58, which isn't correct. Fully hyperbolic linear layers proposed in prior works (https://arxiv.org/abs/2303.15919, https://arxiv.org/abs/2105.14686, https://arxiv.org/abs/2407.01290) operates on the entire Lorentz vector and the compute the time dimension. This brings into question of whether the authors correctly implemented the baselines for their subsequent experiments.
2. The improvement in vision tasks is incredibly marginal in both the main results and the ablations
3. Because the improvement in CIFAR datasets are marginal, the performance of the proposed method is essentially entirely dependent on the genomic dataset.
4. Some of the citations are repeated, which causes confusion

**Questions:**

1. Does the improvement brought by intrinsic Lorentz operations sustain beyond genomic data, since the improvement on the CIFAR datasets is marginal?
2. Could the authors clarify on discrepancy between their definition of LFC and the corresponding papers (as in weakness 1)?
3. One property of LFC is that it encompasses all Lorentz transformations. Can similar properties be proved for PLFC?

---

> ### Author Response · Authors · 2025-11-21
> **Response to Reviewer 98q5 (1/2)**
>
> Thanks for the reviewer's effort and valuable suggestions.
>
>
> > (W1) The authors claimed that LFC "applies Euclidean mappings to spatial components while treating the time-like coordinate separately" on line 58, which isn't correct. Fully hyperbolic linear layers proposed in prior works (https://arxiv.org/abs/2303.15919, https://arxiv.org/abs/2105.14686, https://arxiv.org/abs/2407.01290) operate on the entire Lorentz vector and compute the time dimension. This brings into question whether the authors correctly implemented the baselines for their subsequent experiments.
>
> All three papers indeed operate on the entire Lorentz vector, then extract the space part from the mapping output and accordingly calculate the time part to form the output vector. Our baselines are actually consistent with previous methods. Thanks to the reviewers again, and we fixed the incorrect description in the new revision.
>
> > (W2&W3) The improvement in vision tasks is incredibly marginal in both the main results and the ablations. Because the improvement in CIFAR datasets is marginal, the performance of the proposed method is essentially entirely dependent on the genomic dataset.
>
> Both PLFC and GyroLBN in ILNN are plug-and-play components that can be applied to arbitrary architectures. Therefore, although the improvements on CIFAR-10/100 are numerically small, what we consider more important is that these modules provide consistent gains across all datasets we evaluate on. To further support this point, we additionally evaluated our method on three widely used graph datasets, as shown in Question 1, where replacing the original hyperbolic FC layer with PLFC again leads to consistent improvements.
>
> > (W4) Some of the citations are repeated, which causes confusion
>
> Thanks for the reminder, we fix it in the new versions.

---

> > ### Author Response · Authors · 2025-11-21
> > **Response to Reviewer 98q5 (2/2)**
> >
> > > (Q1) Does the improvement brought by intrinsic Lorentz operations sustain beyond genomic data, since the improvement on the CIFAR datasets is marginal?
> >
> > The improvement brought by intrinsic Lorentz operations sustains beyond genomic data. We evaluated our method on three widely-used graph datasets, which exhibit intricate topological and hierarchical relationships, making them an ideal testbed for evaluating the effectiveness of hyperbolic network(e.g., HGNN, HGCN, HGAT, HAN, HNN++, and Hypformer[1]). We choose the Hypformer (https://arxiv.org/abs/2407.01290) as baseline and by replacing the Linear layer in Hypformer, gain more than 1% average enhancement.
> >
> >
> > | Models          | Airport             | Cora               | PubMed             |
> > |-----------------|---------------------|--------------------|--------------------|
> > | HGNN            | $84.7\pm1.0$        | $77.1\pm0.8$       | $78.3\pm1.2$       |
> > | HGCN            | $89.3\pm1.2$        | $76.5\pm0.6$       | $78.0\pm1.0$       |
> > | HGAT            | $89.6\pm1.0$        | $77.4\pm0.7$       | $78.3\pm1.4$       |
> > | HAN             | $92.9\pm0.6$        | $83.1\pm0.5$       | $79.0\pm0.6$       |
> > | HNN++           | $92.3 \pm 0.3$      | $82.8 \pm 0.6$     | $79.9 \pm 0.4$     |
> > | F-HNN           | $93.0\pm0.7$        | $81.0\pm0.7$       | $77.5\pm0.8$       |
> > | NodeFormer      | $80.2\pm0.6$        | $82.2 \pm 0.9$     | $79.9 \pm 1.0$     |
> > | SGFormer        | $92.9 \pm 0.5$      | $83.2\pm 0.9$      | $80.0\pm0.8$       |
> > | Hypformer [1]   | $95.0 \pm 0.5$      | $85.0\pm0.3$       | $81.3\pm 0.3$      |
> > | Hypformer+PLFC  | $96.03 \pm 0.34$    | $85.68\pm0.19$     | $82.52\pm 0.33$    |
> >
> > [1]: Hypformer: Exploring Efficient Transformer Fully in Hyperbolic Space
> >
> > > (Q2) Could the authors clarify on discrepancy between their definition of LFC and the corresponding papers (as in weakness 1)?
> >
> > All three papers indeed operate on the entire Lorentz vector, then extract the space part from the mapping output and accordingly calculate the time part to form the output vector. Our baselines are actually with the previous method, HCNN. Thanks to the reviewers again, and we fix the incorrect description in the new revision.
> >
> >
> > > (Q3) One property of LFC is that it encompasses all Lorentz transformations. Can similar properties be proved for PLFC?
> >
> >
> > In the Lorentzian space $\mathbb{L}^n$ $(\mathbb{R}^{n+1}, \langle\cdot,\cdot\rangle_L)$, a linear map $\phi:\mathbb{L}^n\to\mathbb{L}^n$ is called a Lorentz transformation if it preserves the Lorentzian inner product, i.e. $\langle Ax, Ay\rangle_L = \langle x,y\rangle_L$ for all $x,y\in\mathbb{R}^{1+n}$. All such maps $A$ form the Lorentz group $O(1,n)$; in this work, we mainly consider its time-oriented subgroup $O^+(1,n)$, where $a_{00} > 0$.
> >
> > By definition, every Lorentz transformation is a linear operator on $\mathbb{L}^n$. In contrast, our PLFC head applies nonlinear functions such as $\sinh(\cdot)$ together, and is therefore not itself a linear mapping and does not contain all of $O^+(1,n)$. Importantly, Lorentz transformations merely change the coordinate system in which the same hyperbolic geometry is represented, without altering Lorentzian distances or the decision boundary. Hence it is neither necessary nor desirable for PLFC to parameterize all Lorentz transformations. For the theoretical benefit of PLFC, a trivial explanation is in Appendix E.2.
> >
> >
> > Finally, thank you for your support.

---

### Official Review · Reviewer_98q5 · 2025-10-27

**Soundness:** 2
**Presentation:** 2
**Contribution:** 2
**Rating:** 4
**Confidence:** 4

**Summary:**

This paper proposes the intrinsic Lorentz neural network (ILNN), which is claimed to fully operate within the Lorentz model. The main architecture is a point-to-hyperplane Lorentz fully connected layer, along with which the paper introduces Lorentz versions of batch normalization (GyroLBN) and dropout, etc. Experiments on CIFAR-10/100, TEB, and GUE datasets show that ILNN achieves competitive results. Ablation studies are also reported to confirm the contributions of PLFC and GryoLBN.

**Strengths:**

1. The paper addresses an important topic of building neural architecture that respects hyperbolic geometry while avoiding the inconsistent mixture of Euclidean and hyperbolic components.
2. The experiments on vision and genomics tasks show potential interdisciplinary applications of hyperbolic neural networks.
3. On TEB and GUE datasets, ILNN achieves significant improvement on several tasks.
4. The paper is clearly structured, with detailed mathematical formulas and reasonable ablation studies.

**Weaknesses:**

1. The paper claims that its method is “Intrinsic Lorentz”, yet provides no rigorous definition of what that means. If “intrinsic” means depending only on manifold operations, then computing directly in Minkowski coordinates is still extrinsic.
2. On CIFAR-10/100, the gain over Euclidean model is very limited (< 0.5%), and most of the selected baseline models are not even as good as the standard Euclidean model…
3. On TEB and GUE, there should be more baseline methods (some from what is reviewed on page 2) than only Euclidean CNN and HCNN. Maybe other tasks of TEB and GUE can be included as well. Actually, other than using CIFAR, TEB and GUE, why not use some datasets that have been widely used and clearly have hierarchical structures?
4. The use of the “gyro-” terminology is conceptually problematic. The review of gyrovector spaces on page 3 is inaccurate: equations (1) and (2) describe standard Riemannian composition rules using exponential and logarithmic maps, not Ungar’s gyrovector algebra. The Lorentz model does not form a gyrovector space in the algebraic sense, unlike the Poincaré ball.
5. Theorem 1 mixes a definition (the formulation of the PLFC layer) and a theorem statement. These should be clearly separated for readability and mathematical rigor.
6. Some reference items appear multiple times. For instance, the paper by Bdeir appeared three times, but they are actually the same one. This is not the only entry. Please carefully check it.

**Questions:**

1. How is “intrinsic” formally defined? How are existing Lorentz-model methods not intrinsic? Why is “intrinsic Lorentz” beneficial? These have to be defined carefully and mathematically.
2. The result of Theorem 1 is of course interesting. But can you formulate a theorem to show the benefit of your specific design, or benefit of using an intrinsic Lorentz design?
3. Can you introduce some stronger baselines for TEB and GUE?
4. Can you test your model on some better-known datasets where hyperbolic space has proved effective?

---

> ### Author Response · Authors · 2025-11-21
> **Response to Reviewer 98q5 (1/4)**
>
> Thanks for the reviewer's effort and professional suggestions.
>
>
> > (W1) The paper claims that its method is “Intrinsic Lorentz”, yet provides no rigorous definition of what that means. If “intrinsic” means depending only on manifold operations, then computing directly in Minkowski coordinates is still extrinsic. On CIFAR-10/100, the gain over the Euclidean model is very limited (< 0.5%), and most of the selected baseline models are not even as good as the standard Euclidean model…
>
> For the “Intrinsic Lorentz”, please see Q1 below. We also extend our method to three widely used graph datasets in Q4. Here I want to emphasize that bringing consistent improvements across all datasets is more important because this is a geometric optimization. Some selected baseline models reported in previous work are indeed not even as good as the standard Euclidean model, but the recent method has already surpassed the Euclidean model.
>
> > (W2&W3) On TEB and GUE, there should be more baseline methods (some from what is reviewed on page 2) than only Euclidean CNN and HCNN. Maybe other tasks of TEB and GUE can be included as well. Actually, other than using CIFAR, TEB, and GUE, why not use some datasets that have been widely used and clearly have hierarchical structures?
>
> We evaluated our method on three widely-used graph datasets as below in Question 4.
>
> > (W3&W4) The use of the “gyro-” terminology is conceptually problematic. The review of gyrovector spaces on page 3 is inaccurate: equations (1) and (2) describe standard Riemannian composition rules using exponential and logarithmic maps, not Ungar’s gyrovector algebra. The Lorentz model does not form a gyrovector space in the algebraic sense, unlike the Poincaré ball.
> Theorem 1 mixes a definition (the formulation of the PLFC layer) and a theorem statement. These should be clearly separated for readability and mathematical rigor
>
> Thanks for the two suggestions; the Lorentz model actually forms a gyrovector space. Following [2], we have supplemented this content in Appendix C.
>
> Meanwhile, we split Theorem 1 into a definition (the formulation of the PLFC layer) and a theorem statement. Thanks again.
>
>
> > (W5) Some reference items appear multiple times. For instance, the paper by Bdeir appeared three times, but they are actually the same one. This is not the only entry. Please carefully check it.
>
> Thanks for the reminder, we will carefully check it and fix it in the new versions.

---

> > ### Author Response · Authors · 2025-11-21
> > **Response to Reviewer 98q5 (2/4)**
> >
> > > (Q1) How is “intrinsic” formally defined? How are existing Lorentz-model methods not intrinsic? Why is “intrinsic Lorentz” beneficial? These have to be defined carefully and mathematically.
> >
> >
> > We appreciate the reviewer’s request for a precise definition. In our paper, “intrinsic Lorentz” refers to using only operations that are *well-defined on the Lorentz model itself*, rather than on its ambient Minkowski space.
> >
> >
> > We define a layer ($F:\mathcal{L}^n_K\to\mathcal{L}^m_K$) **intrinsic** if
> >
> > 1. its input, output, and all intermediate states lie on some (\mathcal{L}^d_K) (they always satisfy $\langle z,z\rangle_L=-1/K,\ z_0>0)$, and
> > 2. it is expressed entirely in terms of the Lorentzian geometry: $\langle\cdot,\cdot\rangle_L$, the induced distance $d_L$, and operators derived from them (exp/log maps, parallel transport, gyroaddition/gyroscaling, Lorentzian centroids, etc.), **without ever using arbitrary Euclidean linear maps on Lorentz vectors in the ambient space**.
> >
> > Under this definition, the previous Lorentz fully connected layer (LFC) used in HCNN is *not* intrinsic. Its update has the form
> > $$\mathbf{y} = \begin{bmatrix}
> > \sqrt{|\phi(W\mathbf{x},\mathbf{v})|^2 - 1/K} ,
> > \phi(W\mathbf{x},\mathbf{v})
> > \end{bmatrix},
> > $$
> > where $\mathbf{x}\in\mathcal{L}^n_K$ and $W\mathbf{x}$ is a standard matrix–vector product in $\mathbb{R}^n$. This $W\mathbf{x}$ is defined using the *ambient* linear structure of the Minkowski space, not any operation on the Lorentz manifold itself. Because the core transformation is an ambient Minkowski multiplication rather than a Lorentzian/geodesic operation, this layer is only partially intrinsic.
> >
> > By contrast, our PLFC is constructed entirely from Lorentz-geometry primitives that have closed-form definitions on $\mathcal{L}^n_K$. Each logit is the signed Lorentzian distance from the input point to a learned Lorentz hyperplane, and the output $\mathbf{y}$ is then recovered in closed form as the unique point on $\mathcal{L}^m_K$ whose signed distances to a set of coordinate hyperplanes equal these logits. All steps (hyperplane parameterization, point-to-hyperplane distance, reconstruction of $\mathbf{y}$) are expressed only via the Lorentzian inner product and distance; no ambient Euclidean affine map $W\mathbf{x}+b$ is ever applied. In this sense, PLFC is an intrinsic Lorentz FC layer.
> >
> > Finally, we emphasize that using **Minkowski coordinates** $x_0,\dots,x_n$ in the implementation does not contradict intrinsicness: coordinates are just a way to write down formulas. What matters is which metric and which operations the formulas use. Our layers always use the Lorentzian inner product and its induced geometry; we never treat $\mathcal{L}^n_K$ as a Euclidean vector space. The non-intrinsic part of prior work is not “writing in Minkowski coordinates,” but **performing ambient Euclidean linear operations on Lorentz vectors** (such as $W\mathbf{x}$) that are not defined by the Lorentz geometry itself.

---

> ### Author Response · Authors · 2025-11-21
> **Response to Reviewer 98q5 (3/4)**
>
> > (Q2) The result of Theorem 1 is, of course, interesting. But can you formulate a theorem to show the benefit of your specific design, orther benefit of using an intrinsic Lorentz design?
>
> Here is a trivial explanation, assuming only one Lorentz FC layer followed by a Lorentz MLR.
>
> **Theorem (Margin preservation and contraction of PLFC and LFC).**
>
> Fix a curvature $K<0$. For any input $x$, let the penultimate layer produce
> $$
> u(x) = (u_1(x), \dots, u_m(x)) \in \mathbb{R}^m,
> $$
> and let $c$ denote the true class. Define the pre-logit margin
> $$
> \Delta(x) := u_c(x) - \max_{j\neq c} u_j(x).
> $$
>
> Consider two Lorentz output-layer designs that use signed geodesic distances from the output point $y(x)$ to the coordinate hyperplanes as logits:
>
> - *PLFC head (intrinsic).* The spatial coordinates are
>   $$
>   y_{s,k}^{\mathrm{PLFC}}(x)
>   = \frac{1}{\sqrt{-K}}\sinh\big(\sqrt{-K}\,u_k(x)\big),
>   $$
>   and the signed Lorentzian distance to the $k$-th coordinate hyperplane is
>   $$
>   d_k^{\mathrm{PLFC}}(x)
>   = \frac{1}{\sqrt{-K}}\operatorname{asinh}\big(\sqrt{-K}\,y_{s,k}^{\mathrm{PLFC}}(x)\big).
>   $$
>
> - *LFC head (extrinsic/linear).* The spatial coordinates are taken directly as
>   $$
>   y_{s,k}^{\mathrm{LFC}}(x) = u_k(x),
>   $$
>   and the signed Lorentzian distance to the same hyperplane is
>   $$
>   d_k^{\mathrm{LFC}}(x)
>   = \frac{1}{\sqrt{-K}}\operatorname{asinh}\big(\sqrt{-K}\,u_k(x)\big).
>   $$
>
> Define the distance-based margins
> $$
> \Delta^{\mathrm{PLFC}}(x) := d_c^{\mathrm{PLFC}}(x) - \max_{j\neq c} d_j^{\mathrm{PLFC}}(x),
> $$
> $$
> \Delta^{\mathrm{LFC}}(x) := d_c^{\mathrm{LFC}}(x) - \max_{j\neq c} d_j^{\mathrm{LFC}}(x).
> $$
>
> Then, for every sample $x$:
>
> 1. **(Margin preservation of PLFC)**
>    $$
>    \Delta^{\mathrm{PLFC}}(x) = \Delta(x).
>    $$
>
> 2. **(Margin contraction of LFC)**
>    The LFC head preserves the sign of the margin and contracts its magnitude:
>    $$
>    \operatorname{sign}\big(\Delta^{\mathrm{LFC}}(x)\big)
>    = \operatorname{sign}\big(\Delta(x)\big),
>    $$
>    $$
>    \bigl|\Delta^{\mathrm{LFC}}(x)\bigr|
>    \le \bigl|\Delta(x)\bigr|,
>    $$
>    with strict inequality $\bigl|\Delta^{\mathrm{LFC}}(x)\bigr| < \bigl|\Delta(x)\bigr|$ whenever $\Delta(x)\neq 0$.
>
>
> Proof is in Appendix E.2. These two properties show that the intrinsic PLFC head preserves and exposes the useful discriminative information in the pre-logits, while the extrinsic LFC head can only shrink it. The identity relation $\Delta^{\mathrm{PLFC}}(x) = \Delta(x)$ means that any margin gained by the backbone is passed unchanged to the decision layer: the hyperbolic geometry of the PLFC head does not distort class separation and thus fully preserves large-margin behavior (and its usual benefits for robustness and calibration). In contrast, the sign-preserving contraction $\bigl|\Delta^{\mathrm{LFC}}(x)\bigr| \le \bigl|\Delta(x)\bigr|$ shows that the LFC head cannot increase the absolute margin and typically reduces it, so correctly classified points become less well separated and the classifier operates with systematically smaller effective margins. Together, this explains why, under the same backbone features, an intrinsic Lorentz design like PLFC provides a strictly more faithful and margin-friendly interface between Euclidean pre-logits and hyperbolic decision boundaries than an extrinsic, linear Lorentz head.

---

> ### Author Response · Authors · 2025-11-21
> **Response to Reviewer 98q5 (4/4)**
>
> > (Q3) Can you introduce some stronger baselines for TEB and GUE?
>
> To the best of our knowledge, at the time of submission (before September), there were no stronger published baselines on TEB and GUE beyond the CNN/HCNN family introduced in “Hyperbolic Genome Embeddings”. We therefore adopted exactly those models (Euclidean CNN, HCNN-S/M) as the main reference points on these benchmarks.
>
>
> More importantly, our main contribution (PLFC + GyroLBN) is architecture-agnostic: PLFC is a plug-and-play, intrinsic Lorentz fully connected layer that can replace existing hyperbolic FC heads in other models. Even when we move beyond CNN-based designs—for example, in a forthcoming transformer-based hyperbolic architecture (Hypformer) appearing in Q4—replacing its original FC with PLFC still yields consistent improvements. This suggests that, if stronger hyperbolic baselines for genomics appear in the future, our modules can be directly integrated into them as well.
>
>
> > (Q4) Can you test your model on some better-known datasets where hyperbolic space has proved effective?
>
> We evaluated our method on three widely-used graph datasets, which exhibit intricate topological and hierarchical relationships, making them an ideal testbed for evaluating the effectiveness of hyperbolic network(e.g, HGNN, HGCN, HGAT, HAN, HNN++, and Hypformer[1]). We choose the Hypformer as baseline, and by replacing the Linear layer in Hypformer, we gain more than 1% average enhancement.
>
>
> | Models          | Airport             | Cora               | PubMed             |
> |-----------------|---------------------|--------------------|--------------------|
> | HGNN            | $84.7\pm1.0$        | $77.1\pm0.8$       | $78.3\pm1.2$       |
> | HGCN            | $89.3\pm1.2$        | $76.5\pm0.6$       | $78.0\pm1.0$       |
> | HGAT            | $89.6\pm1.0$        | $77.4\pm0.7$       | $78.3\pm1.4$       |
> | HAN             | $92.9\pm0.6$        | $83.1\pm0.5$       | $79.0\pm0.6$       |
> | HNN++           | $92.3 \pm 0.3$      | $82.8 \pm 0.6$     | $79.9 \pm 0.4$     |
> | F-HNN           | $93.0\pm0.7$        | $81.0\pm0.7$       | $77.5\pm0.8$       |
> | NodeFormer      | $80.2\pm0.6$        | $82.2 \pm 0.9$     | $79.9 \pm 1.0$     |
> | SGFormer        | $92.9 \pm 0.5$      | $83.2\pm 0.9$      | $80.0\pm0.8$       |
> | Hypformer [1]   | $95.0 \pm 0.5$      | $85.0\pm0.3$       | $81.3\pm 0.3$      |
> | Hypformer+PLFC  | $96.03 \pm 0.34$    | $85.68\pm0.19$     | $82.52\pm 0.33$    |
>
> [1]: Hypformer: Exploring Efficient Transformer Fully in Hyperbolic Space. KDD24
> [2]: Riemannian Batch Normalization: A Gyro Approach.
>
> The revisions mentioned above are all presented in the revised manuscript; we sincerely hope you will reconsider our rate. Thanks.

---

### Official Review · Reviewer_9oRv · 2025-11-01

**Soundness:** 3
**Presentation:** 3
**Contribution:** 3
**Rating:** 6
**Confidence:** 4

**Summary:**

This paper proposes Intrinsic Lorentz Neural Network which is a fully intrinsic hyperbolic architecture that conducts all computations within the Lorentz model. The paper introduces several intrinsic modules including a novel point-to-hyperplane fully connected layer and GyroLBN. Several other components including a gyroadditive bias for the FC output, a Lorentz patch-concatenation operator and a Lorentz dropout layer are also introduced. Extensive experiments conducted on CIFAR-10/100 and two genomic benchmarks (TEB and GUE) illustrate that ILNN achieves state-of-the-art performance.

**Strengths:**

1. The proposed Intrinsic Lorentz Neural Network is well motivated as existing hyperbolic architectures remain partially intrinsic, mixing Euclidean operations with hyperbolic ones or relying on extrinsic parameterizations.

2. The proposed point-to-hyperplane Lorentz fully connected layer is new in the literature of hyperbolic neural network which replaces traditional affine transformations with intrinsic hyperbolic distance.

3. The paper is also generally well-written and extensive experimental results show the improvements of the proposed Intrinsic Lorentz Neural Network.

**Weaknesses:**

1. The experiments are only conducted on CIFAR-10/100  and two genomic benchmarks, it would be interesting to show if the proposed Intrinsic Lorentz Neural Network can outperform existing hyperbolic neural networks on large-scale datasets such as ImageNet and datasets with long-tail class distributions.

2. The authors argue that intrinsic designs are preferable compared with partially intrinsic or extrinsic parameterizations, but there is a lack of enough evidence showing that the claim is true. There are quantitative results but the performance of ILNN is very similar to that of Hybrid Lorentz. Therefore, it is not clear if such intrinsic designs can bring significant benefits.

3. In table 3, the authors analyze training efficiency to disentangle the effects of the classifier design and the normalization strategy, but there is no comparison of training time of the proposed Intrinsic Lorentz Neural Network with other hyperbolic neural networks mentioned in table 1.

**Questions:**

Besides the weaknesses, I have the following additional questions,

1. The authors use CIFAR-10 and CIFAR-100 in the experiments to demonstrate the benefits of the intrinsic approach. However, it is unclear what kind of hierarchical structure these two datasets possess. Therefore, it is difficult to conclude that the intrinsic approach preserves the geometry of the Lorentz model without distortion. Could the authors consider using datasets with explicit hierarchical structures to better support their claim?

2. The authors mentioned fully connected layer and convolutional layer, I am wondering if the same intrinsic approach can be applied to build other kind of neural network layers such as attention layers？

---

> ### Author Response · Authors · 2025-11-21
> **Response to Reviewer 9oRv (1/3)**
>
> > (W1) The experiments are only conducted on CIFAR-10/100 and two genomic benchmarks; it would be interesting to show if the proposed Intrinsic Lorentz Neural Network can outperform existing hyperbolic neural networks on large-scale datasets such as ImageNet and datasets with long-tail class distributions.
>
> For “large-scale” evaluation, we note that our experiments are not restricted to tiny datasets: within the GUE benchmark, the Covid-variant classification task already contains on the order of **10,000 sequences**, which is considered large-scale in genomics. On this dataset, ILNN not only avoids the collapse of existing hyperbolic CNNs but also, firstly, surpasses the Euclidean CNN, showing that the intrinsic Lorentz design remains effective beyond small toy problems. Training a fully hyperbolic model on ImageNet is extremely time-consuming, so we are trying, but this does not influence the result for the large-scale Covid dataset supports the effectiveness of our method.
>
> As for long-tail, the introductory statement that “real-world data frequently exhibit latent hierarchical structures and long-tail distributions, which can be naturally represented by hyperbolic geometry” is intended as a high-level summary of a now well-established line of work: a large body of prior research has shown that hyperbolic representations are particularly effective for data with latent hierarchies [1][2][3] and long-tailed [4][5][6] structure. In this sense, choosing a negatively curved space for such domains is no longer a speculative design choice but a natural and increasingly standard modeling paradigm. Building on this general motivation, our contribution is not to re-argue for hyperbolic geometry itself, but to design an **intrinsic Lorentz architecture** that respects the underlying curvature at every layer (classifier, bias, normalization, convolution, concatenation, dropout, activation), thereby better exploiting the advantages of hyperbolic geometry, which preserve hierarchical structure more faithfully throughout the network. So we mainly focus on the hierarchical structure and fix the introductory statement to make our motivation clearer. Thanks for the suggestion again.
>
>
> > (W2) The authors argue that intrinsic designs are preferable compared with partially intrinsic or extrinsic parameterizations, but there is a lack of sufficient evidence showing that the claim is true. There are quantitative results, but the performance of ILNN is very similar to that of Hybrid Lorentz. Therefore, it is not clear if such intrinsic designs can bring significant benefits.
>
> Both PLFC and GyroLBN in ILNN are plug-and-play components that can be applied to arbitrary architectures. Therefore, although the improvements on CIFAR-10/100 are numerically small, what we consider more important is that these modules provide consistent gains across all datasets we evaluate on. To further support this point, we additionally evaluated our method on three widely used graph datasets, as shown in Question 1, where replacing the original hyperbolic FC layer with PLFC again leads to consistent improvements.
>
>
> > (W3) In Table 3, the authors analyze training efficiency to disentangle the effects of the classifier design and the normalization strategy, but there is no comparison of the training time of the proposed Intrinsic Lorentz Neural Network with other hyperbolic neural networks mentioned in Table 1.
>
> Most methods in Table 1 (except HCNN) are essentially Euclidean ResNet-18 backbones with only a hyperbolic MLR head, so they are very close to standard Euclidean CNNs in terms of runtime(12s per epoch). In contrast, fully hyperbolic models (including ours and HCNN) necessarily incur extra cost from richer hyperbolic operations, so they are expected to be slower than a purely Euclidean ResNet, there is no free lunch if we want the geometric benefits. For this reason, Table 3 focuses on fair timing comparisons within the hyperbolic family, and shows that ILNN with PLFC and especially GyroLBN is already more efficient than previous Lorentz CNN variants (e.g., HCNN) while also improving accuracy. A systematic, implementation-optimized speed comparison against all Euclidean-style baselines is an interesting direction, but lies beyond the main scope of this work.

---

> > ### Author Response · Authors · 2025-11-21
> > **Response to Reviewer 9oRv (2/3)**
> >
> > > (Q1) The authors use CIFAR-10 and CIFAR-100 in the experiments to demonstrate the benefits of the intrinsic approach. However, it is unclear what kind of hierarchical structure these two datasets possess. Therefore, it is difficult to conclude that the intrinsic approach preserves the geometry of the Lorentz model without distortion. Could the authors consider using datasets with explicit hierarchical structures to better support their claim?
> >
> >
> > 1. The benchmarks we evaluate on are all drawn from datasets with clear hierarchical structure and high hyperbolicity (low $δ_{rel}$ [1][2]) as shown in the table, which are also used and verified in paper [1][2]. Detailed descriptions are as follows, and we revise them to the appendix.
> >     **CIFAR-10/100.**
> >     CIFAR-10 and CIFAR-100 are standard proxies for visual object recognition whose categories naturally admit semantic hierarchies (e.g., animal → mammal → dog → specific breed). In CIFAR-100, this is made explicit by grouping the 100 fine-grained classes into 20 coarse superclasses in the original dataset design. This **hierarchical structure** has been extensively verified in prior hyperbolic vision work [1][2] on CIFAR-10/100, and even [1] claimed that ``CIFAR-10/100 exhibit hierarchical class relations and high hyperbolicity, making the use of hyperbolic models well-motivated``. Our use of CIFAR is therefore to test whether the proposed intrinsic Lorentz geometry remains competitive on widely used vision benchmarks with well-defined hierarchical label taxonomies.
> >
> >     **TEB and GUE**
> >     The TEB and GUE suites are constructed directly from *natural genomic sequences* and inherit the biological hierarchies of their domains. TEB tasks span multiple **transposable-element families** across retrotransposons, DNA transposons, and pseudogenes, which themselves sit in a multi-level taxonomic hierarchy (orders → superfamilies → families). GUE aggregates datasets for transcription-factor binding, promoter and core-promoter detection, splice-site prediction, and COVID-variant classification across several species. These tasks are all manifestations of hierarchical regulatory structure (e.g., motifs → modules → promoters → gene expression).
> >
> > | Benchmark | Task                     | Dataset  | δ_rel |
> > |-----------|--------------------------|----------|-------|
> > | CIFAR     | Image classification     | CIFAR-10 | 0.26  |
> > | CIFAR     | Image classification     | CIFAR-100| 0.23  |
> > | TEB       | Pseudogenes              | processed| 0.19  |
> > | TEB       | Pseudogenes              | unprocessed | 0.16 |
> > | GUE       | Covid variant classification | covid | 0.42  |
> > | GUE       | Core Promoter detection  | all      | 0.23  |
> > | GUE       | Core Promoter detection  | notata   | 0.21  |
> > | GUE       | Core Promoter detection  | tata     | 0.14  |
> > | GUE       | Promoter detection       | all      | 0.26  |
> > | GUE       | Promoter detection       | notata   | 0.26  |
> > | GUE       | Promoter detection       | tata     | 0.14  |
> >
> > Furthermore, we evaluated our method on three widely-used graph datasets, which exhibit intricate topological and hierarchical relationships, making them an ideal testbed for evaluating the effectiveness of hyperbolic network(e.g., HGNN, HGCN, HGAT, HAN, HNN++, and Hypformer[1]). We choose the Hypformer as baseline, and by replacing the Linear layer in Hypformer, we gain more than 1% average enhancement.
> >
> > | Models          | Airport             | Cora               | PubMed             |
> > |-----------------|---------------------|--------------------|--------------------|
> > | HGNN            | $84.7\pm1.0$        | $77.1\pm0.8$       | $78.3\pm1.2$       |
> > | HGCN            | $89.3\pm1.2$        | $76.5\pm0.6$       | $78.0\pm1.0$       |
> > | HGAT            | $89.6\pm1.0$        | $77.4\pm0.7$       | $78.3\pm1.4$       |
> > | HAN             | $92.9\pm0.6$        | $83.1\pm0.5$       | $79.0\pm0.6$       |
> > | HNN++           | $92.3 \pm 0.3$      | $82.8 \pm 0.6$     | $79.9 \pm 0.4$     |
> > | F-HNN           | $93.0\pm0.7$        | $81.0\pm0.7$       | $77.5\pm0.8$       |
> > | NodeFormer      | $80.2\pm0.6$        | $82.2 \pm 0.9$     | $79.9 \pm 1.0$     |
> > | SGFormer        | $92.9 \pm 0.5$      | $83.2\pm 0.9$      | $80.0\pm0.8$       |
> > | Hypformer [7]   | $95.0 \pm 0.5$      | $85.0\pm0.3$       | $81.3\pm 0.3$      |
> > | Hypformer+PLFC  | $96.03 \pm 0.34$    | $85.68\pm0.19$     | $82.52\pm 0.33$    |

---

> ### Author Response · Authors · 2025-11-21
> **Response to Reviewer 9oRv (3/3)**
>
> > (Q2) The authors mentioned a fully connected layer and a convolutional layer. I am wondering if the same intrinsic approach can be applied to build other kinds of neural network layers, such as attention layers？
>
> The intrinsic approach surely can be applied to build other kinds of neural network layers. The attention mechanism consists of three components: transformation, similarity, and aggregation. Here, we replace the transformation (a Linear layer) in Hypformer[7] by our PLFC and gain consistent improvement across all three datasets, as shown in the table containing Hypformer above.
>
>
> Finally, thank you for your support.
>
>
> \[1]: Fully Hyperbolic Convolutional Neural Networks for Computer Vision. ICLR2024
> [2]: Neural networks on Symmetric Spaces of Noncompact Type. ICLR2025
> [3]: Hyperbolic Genome Embeddings. ICLR2025
> [4]: Hyperbolic Hypergraphs for Sequential Recommendation. CIKM2021
> [5]: Curvature Learning for Generalization of Hyperbolic Neural Networks. IJCV2025
> [6]: Knowledge Graph Completion Method based on Hyperbolic Representation Learning and Contrastive Learning. EIJ2023
> [7]: Hypformer: Exploring Efficient Transformer Fully in Hyperbolic Space

---

### Official Review · Reviewer_qw2r · 2025-11-01

**Soundness:** 2
**Presentation:** 3
**Contribution:** 2
**Rating:** 2
**Confidence:** 4

**Summary:**

This work proposes to learn representation and classification with all computations in the Lorentz model for hierarchical and long-tail data using point-to-hyperplane layers and intrinsic normalization, tested on image and genomic datasets.

**Strengths:**

This paper's main contribution proposes a new point-to-hyperplane fully connected layer (PLFC) that utilizes intrinsic Lorentzian distance for logits to match the data's latent hierarchy. However, this approach is not originally developed by the authors and lacks novelty. Furthermore, the mechanism by which this method achieves matching with the data's latent hierarchy remains unclear and insufficiently explained.

**Weaknesses:**

(1) The paper claims to focus on real-world data with latent hierarchical structure and long-tail distribution. However, I did not find clear evidence that CIFAR-10, CIFAR-100, or the TEB and GUE genomics benchmarks possess such properties.

(2) The PLFC method referenced by the authors as originating from HNN++ had been previously proposed in works on fully hyperbolic neural networks and the hypformer paper ("Hypformer: Exploring Efficient Transformer Fully in Hyperbolic Space"), yet the authors do not cite or compare with these relevant methods.

(3) Most other modules in the paper appear to be adopted from existing literature; I do not observe any genuine innovation.

(4) It remains unclear to me how the proposed method specifically addresses latent hierarchical structure and long-tail distribution in the data.

**Questions:**

In the datasets mentioned, where exactly do you observe the latent hierarchical structure and long-tail distribution? Could you explain why hyperbolic methods are effective for addressing latent hierarchical structures and long-tail distributions? Additionally, what specifically constitutes the hierarchical structure in this context?

---

> ### Author Response · Authors · 2025-11-21
> **Response to Reviewer qw2r (1/4)**
>
> Thanks for the reviewer's effort.
> > (W1) The paper claims to focus on real-world data with a latent hierarchical structure and long-tail distribution. However, I did not find clear evidence that CIFAR-10, CIFAR-100, or the TEB and GUE genomics benchmarks possess such properties.
>
> The introductory statement that “real-world data frequently exhibit latent hierarchical structures and long-tail distributions, which can be naturally represented by hyperbolic geometry” is intended as a high-level summary of a now well-established line of work: a large body of prior research has shown that hyperbolic representations are particularly effective for data with latent hierarchies [1][2][3] and long-tailed [4][5][6] structure. In this sense, choosing a negatively curved space for such domains is no longer a speculative design choice but a natural and increasingly standard modeling paradigm. Building on this general motivation, our contribution is not to re-argue for hyperbolic geometry itself, but to design an **intrinsic Lorentz architecture** that respects the underlying curvature at every layer (classifier, bias, normalization, convolution, concatenation, dropout, activation), thereby better exploiting the advantages of hyperbolic geometry, which preserve hierarchical structure more faithfully throughout the network. The benchmarks we evaluate on are all drawn from datasets with a clear hierarchical structure:
>
> **CIFAR-10/100.**
> CIFAR-10 and CIFAR-100 are standard proxies for visual object recognition whose categories naturally admit semantic hierarchies (e.g., animal → mammal → dog → specific breed). In CIFAR-100, this is made explicit by grouping the 100 fine-grained classes into 20 coarse superclasses in the original dataset design. This **hierarchical structure** has been extensively verified in prior hyperbolic vision work [1][2] on CIFAR-10/100, and even [1] claimed that ``CIFAR-10/100 exhibit hierarchical class relations and high hyperbolicity, making the use of hyperbolic models well-motivated``. Our use of CIFAR is therefore to test whether the proposed intrinsic Lorentz geometry remains competitive on widely used vision benchmarks with well-defined hierarchical label taxonomies.
>
> **TEB and GUE**
> The TEB and GUE suites are constructed directly from *natural genomic sequences* and inherit the biological hierarchies of their domains. TEB tasks span multiple **transposable-element families** across retrotransposons, DNA transposons, and pseudogenes, which themselves sit in a multi-level taxonomic hierarchy (orders → superfamilies → families). GUE aggregates datasets for transcription-factor binding, promoter and core-promoter detection, splice-site prediction, and COVID-variant classification across several species. These tasks are all manifestations of hierarchical regulatory structure (e.g., motifs → modules → promoters → gene expression).
>
> Furthermore, all these datasets exhibit hierarchical class relations and high hyperbolicity (low $δ_{rel}$[1][2]) as shown in the table, making the use of hyperbolic models well-motivated. We will incorporate these clarifications and dataset descriptions into the revised appendix manuscript to make our motivation more explicit.
>
> | Benchmark | Task                     | Dataset  | δ_rel |
> |-----------|--------------------------|----------|-------|
> | CIFAR     | Image classification     | CIFAR-10 | 0.26  |
> | CIFAR     | Image classification     | CIFAR-100| 0.23  |
> | TEB       | Pseudogenes              | processed| 0.19  |
> | TEB       | Pseudogenes              | unprocessed | 0.16 |
> | GUE       | Covid variant classification | covid | 0.42  |
> | GUE       | Core Promoter detection  | all      | 0.23  |
> | GUE       | Core Promoter detection  | notata   | 0.21  |
> | GUE       | Core Promoter detection  | tata     | 0.14  |
> | GUE       | Promoter detection       | all      | 0.26  |
> | GUE       | Promoter detection       | notata   | 0.26  |
> | GUE       | Promoter detection       | tata     | 0.14  |

---

> ### Author Response · Authors · 2025-11-21
> **Response to Reviewer qw2r (2/4)**
>
> > (W2) The PLFC method referenced by the authors as originating from HNN++ had been previously proposed in works on fully hyperbolic neural networks and the hypformer paper ("Hypformer: Exploring Efficient Transformer Fully in Hyperbolic Space"), yet the authors do not cite or compare with these relevant methods.
>
>
> 1. The baseline HCNN, which we use in our experiments, already incorporates the FC layer from "fully hyperbolic neural networks"(F-HNN) as its FC component, except that HCNN removes the original FC normalization operation, which HCNN's appendix proves to be beneficial. Hypformer’s FC design is the same as HCNN, only adding curvature transitions between layers. But our network keeps curvature unchanged, so we are **already comparing against the F-HNN and hyperformer's FC** through the HCNN baseline.
>
> | Methon| CIFAR-100|
> |-------|----------|
> |FC with norm (F-HNN) |$76.98\pm0.18$|
> |FC without norm (HCNN)| $78.07\pm0.21$ |
> |PLFC|  $78.41\pm0.23$|
>
>
> 2. **Our PLFC totally differs from all three in both formulation (HCNN, F-HNN, and Hyperformer) and role in the overall architecture.** PLFC is constructed directly from the Lorentz hyperplane parameterization and closed-form point-to-hyperplane distance, with a geometric guarantee that each logit equals a signed Lorentz distance and that the output lies exactly on the upper sheet of the hyperboloid, without any ad-hoc projection. This allows us to obtain an **entirely intrinsic Lorentz CNN** (ILNN) and enhance the performance.
>
>     To further support that PLFC is not just a rebranding of an existing FC, but a **drop-in intrinsic alternative** that can benefit other architectures, we additionally evaluated replacing the FC layer in Hypformer with PLFC while keeping all other components unchanged, including the curvature transitions between layers. On the node classification benchmarks used in the Hypformer paper, we obtain consistent improvements:
>
> | Models          | Airport             | Cora               | PubMed             |
> |-----------------|---------------------|--------------------|--------------------|
> | HGNN            | $84.7\pm1.0$        | $77.1\pm0.8$       | $78.3\pm1.2$       |
> | HGCN            | $89.3\pm1.2$        | $76.5\pm0.6$       | $78.0\pm1.0$       |
> | HGAT            | $89.6\pm1.0$        | $77.4\pm0.7$       | $78.3\pm1.4$       |
> | HAN             | $92.9\pm0.6$        | $83.1\pm0.5$       | $79.0\pm0.6$       |
> | HNN++           | $92.3 \pm 0.3$      | $82.8 \pm 0.6$     | $79.9 \pm 0.4$     |
> | F-HNN           | $93.0\pm0.7$        | $81.0\pm0.7$       | $77.5\pm0.8$       |
> | NodeFormer      | $80.2\pm0.6$        | $82.2 \pm 0.9$     | $79.9 \pm 1.0$     |
> | SGFormer        | $92.9 \pm 0.5$      | $83.2\pm 0.9$      | $80.0\pm0.8$       |
> | Hypformer [7]   | $95.0 \pm 0.5$      | $85.0\pm0.3$       | $81.3\pm 0.3$      |
> | Hypformer+PLFC  | $\mathbf{96.03 \pm 0.34}$ | $\mathbf{85.68\pm0.19}$ | $\mathbf{82.52\pm 0.33}$ |
>
> These results show that PLFC differs from HCNN, F-HNN, and Hyperformer. We will report these experiments and revise the citation of HNN++ / F-HNN / Hypformer in the updated manuscript.

---

> > ### Author Response · Authors · 2025-11-21
> > **Response to Reviewer qw2r (3/4)**
> >
> > > (W3) Most other modules in the paper appear to be adopted from existing literature; I do not observe any genuine innovation.
> >
> > As summarized in the introduction, the paper’s technical novelty is not just in using previously known Lorentz primitives, but in (i) a new Lorentz point-to-hyperplane FC layer and (ii) a new batch normalization scheme, GyroLBN, (iii) together with other modules into a fully intrinsic Lorentz CNN.
> >
> > 1. PLFC. As discussed in our response to W2, the proposed Point-to-hyperplane Lorentz Fully Connected (PLFC) layer is not equivalent to the FC layers used in HCNN, F-HNN, and Hypformer. It is derived directly from the Lorentz hyperplane parameterization and the closed-form point-to-hyperplane distance.
> > 2. GyroLBN: a new normalization layer, not just reused GyroBN/LBN.
> >
> >     Conceptually: GyroLBN unifies two previously ideas: the Lorentzian centroid–based statistics of LBN (fast, closed-form mean in Lorentz space), and the gyro-centering + variance-controlled gyro-scaling paradigm of GyroBN (which aligns distributions under gyro-operations but relies on Fréchet means). Concretely, GyroLBN keeps the gyrogroup normalization map (gyro-centering, gyro-scaling, gyro-bias) of GyroBN, but replaces its iterative Fréchet mean with closed-form Lorentzian centroids and dispersion. This design **preserves the geometric behavior of GyroBN while removing its main computational bottleneck**.
> >
> >     Generic Riemannian BN and GyroBN require solving Fréchet mean problems iteratively, which is costly for large batches and 2D convolutions. In GyroLBN, both the batch mean and the dispersion are computed in closed form: one centroid computation and one pass of squared Lorentz distances, with no inner optimization loop.
> > Section 5.3 and Table 3 show that GyroLBN provides consistent accuracy improvements over both LBN and GyroBN across CIFAR-10 and genomics tasks, while also being substantially faster. For example, on CIFAR-10 with PLFC, GyroLBN improves accuracy over PLFC+GyroBN and reduces per-epoch training time from **314 s to 169 s (≈46% reduction)**. Table 5 (appendix) further shows that under matched iteration budgets, GyroLBN remains the best-performing normalization scheme on GUE. Together, these results indicate that GyroLBN is a genuinely new and practically important normalization layer.
> >
> > 3. PLFC, together with Log-radius concatenation, forms a new Lorentz convolution module. And all module jointly forms our intrinsic Lorentz neural network, and gain SOTA performance across all datasets.
> >
> >
> > > (W4) It remains unclear to me how the proposed method specifically addresses latent hierarchical structure and long-tail distribution in the data.
> >
> > Prior work has established [1][2][3][4][5][6] that negatively curved (hyperbolic) spaces are a natural fit for data with latent hierarchies and long-tail statistics: the exponential volume growth allows low-distortion embeddings of tree-like structure, while the radial coordinate naturally captures frequency/specificity (common concepts near the origin, rare/long-tail ones at larger radius). Our contribution is to use this geometry intrinsically at every step, so those advantages are not diluted by Euclidean detours: PLFC defines logits as signed Lorentz distances to learned hyperplanes (curvature-aware margins that align with geodesics), and GyroLBN computes closed-form Lorentzian statistics with gyro-centering/scaling, preserving the semantic meaning of radius and angles under imbalance. Combined in our fully intrinsic Lorentz CNN, the representation, normalization, and decision boundaries all live in hyperbolic space, enabling the model to better maintain hierarchical relations and allocate capacity to long-tail regions—reflected by consistent gains across vision and genomics benchmarks.

---

> ### Author Response · Authors · 2025-11-21
> **Response to Reviewer qw2r (4/4)**
>
> > (Q1) In the datasets mentioned, where exactly do you observe the latent hierarchical structure and long-tail distribution?
>
> As detailed in our response to W1, for CIFAR-10/100, the classes come from natural image recognition where categories are embedded in clear semantic hierarchies (e.g., animal → mammal → dog → specific breed), and CIFAR-100 makes this explicit via its 20 coarse / 100 fine labels. For TEB and GUE, the data are natural genomic sequences with inherent biological hierarchies: transposable elements are organized into orders → superfamilies → families, and regulatory tasks (TF binding, promoters, splice sites, Covid variants) sit in a hierarchy from short motifs to regulatory modules to gene-level function. Furthermore, all these datasets exhibit hierarchical class relations and high hyperbolicity (low $δ_{rel}$) as shown in the table, making the use of hyperbolic models well-motivated.
>
> > (Q2) Could you explain why hyperbolic methods are effective for addressing latent hierarchical structures and long-tail distributions?
>
> As detailed in our related-work discussion, prior work [1][2][3][4][5][6] has shown that negatively curved (hyperbolic) spaces are well suited to data with latent hierarchies and long tails because their volume grows exponentially with radius, which matches the exponential branching of trees: hierarchical structures can be embedded with much lower distortion than in Euclidean space, where one would need very high dimensions to achieve the same. In such hyperbolic embeddings, the radial coordinate naturally correlates with depth/specificity/frequency—coarse or frequent concepts stay near the origin, while fine-grained or rare (long-tail) concepts live further out, while angular directions capture semantic variation at a given level. Hyperbolic methods, therefore, provide a geometry where both hierarchical relations (via depth) and long-tail structure (via radius/frequency) can be represented and separated using relatively simple distance-based operations, which is exactly what our intrinsic Lorentz layers (PLFC and GyroLBN) are designed to exploit.
>
>
> > (Q3) Additionally, what specifically constitutes the hierarchical structure in this context?
>
> In our context, hierarchical structure means that examples and labels are naturally organized in multiple semantic or biological levels rather than being flat and unrelated. For images, this is the class taxonomy (e.g., object → superclass → fine-grained class, such as animal → mammal → dog → specific breed). For genomics, it is the biological hierarchy (e.g., TE order → superfamily → family, or motif → regulatory module → promoter/gene function, or lineage → variant → sub-variant). Our methods aim to preserve and exploit these multi-level parent–child / coarse–fine relationships in the representation space.
>
>
>
> \[1]: Fully Hyperbolic Convolutional Neural Networks for Computer Vision. ICLR2024
>
> [2]: Neural networks on Symmetric Spaces of Noncompact Type. ICLR2025
>
> [3]: Hyperbolic Genome Embeddings. ICLR2025
>
> [4]: Hyperbolic Hypergraphs for Sequential Recommendation. CIKM2021
>
> [5]: Curvature Learning for Generalization of Hyperbolic Neural Networks. IJCV2025
>
> [6]: Knowledge Graph Completion Method based on Hyperbolic Representation Learning and Contrastive Learning. EIJ2023
>
> [7]: Hypformer: Exploring Efficient Transformer Fully in Hyperbolic Space
>
>
> The revisions mentioned above are all presented in the revised manuscript; we sincerely hope you will reconsider our rate. Thanks.

---

> ### Author Response · Authors · 2025-12-02
> **More intuitive differences between PLFC and LFC in their formulas**
>
> LFC was introduced in *Fully Hyperbolic Neural Networks* and then reused (with minor variants) in HCNN and Hypformer; PLFC is a different construction from all of them. This is easiest to see by writing down the four formulas.
>
> 1. **LFC in Fully Hyperbolic Neural Networks (original form)**
>    $$
>    y =
>    \begin{bmatrix}
>    \sqrt{\|\phi(Wx, v)\|_2^2 - 1/K}, \\
>    \phi(Wx, v)
>    \end{bmatrix},
>    $$
>    where $\phi$ denotes dropout / activation / normalization applied to the Euclidean transform $Wx$.
>
> 2. **LFC used in HCNN (dropping $\phi$ in practice)**
>
>    In HCNN the authors explicitly report in the supplement that using $\phi(Wx,v) = Wx + b$ works better in practice, giving
>    $$
>    y =
>    \begin{bmatrix}
>    \sqrt{\|Wx+b\|_2^2 - 1/K}, \\
>    Wx+b
>    \end{bmatrix}.
>    $$
>    This is the LFC variant we use as the baseline in our experiments.
>
> 3. **Hypformer “HTC” layer is just LFC/HCNN with curvature conversion**
>
>    Hypformer introduces a curvature–conversion step
>    $$
>    \operatorname{HTC}(x; f_t, W, \kappa_1, \kappa_2)
>    :=
>    \begin{bmatrix}
>    \sqrt{\frac{\kappa_1}{\kappa_2}\,\|f_t(x; W)\|_2^2 - 1/\kappa_2}, \\
>    \sqrt{\frac{\kappa_1}{\kappa_2}}\, f_t(x; W)
>    \end{bmatrix}.
>    $$
>    where $f_t(x; W)= Wx+b$
>    In the setting the curvature is fixed inside the network ($\kappa_1 = \kappa_2$), so HTC reduces exactly to the LFC in HCNN with $Wx+b$ replaced by $f_t(x;W)$. **Hypformer does not introduce a new LFC; it reuses the same LFC structure already proposed in Fully Hyperbolic Neural Networks and HCNN.** Since we already compared against the HCNN with LFC, we are effectively also comparing against the LFC used in Hypformer.
>
> 4. **Our PLFC is structurally different from LFC/HCNN/Hypformer**
>
>    First, the Lorentz MLR gives the signed point–to–hyperplane distance
>
>    $$
>    v_k(x) = v_{z_k,a_k}(x)=\frac{1}{\sqrt{-K}}\,\operatorname{sign}(\alpha_k)\,\beta_k\,\left|\sinh^{-1}\left( \sqrt{-K}\,\frac{\alpha_k}{\beta_k}\right)\right|,
>    $$
>
>    where (writing $x = (x_t, x_s)$)
>
>    $$
>    \alpha_k= \cosh(\sqrt{-K}\,a_k)\,\langle z_k, x_s\rangle- \sinh(\sqrt{-K}\,a_k)\,\|z_k\|_2\, x_t,
>    $$
>
>    $$
>    \beta_k=\sqrt{\big\|\cosh(\sqrt{-K}\,a_k)\, z_k\big\|_2^2-\big(\sinh(\sqrt{-K}\,a_k)\,\|z_k\|_2\big)^2}.
>    $$
>
>    The PLFC spatial coordinate is then
>    $$
>    y_{s,k}=\frac{1}{\sqrt{-K}}\sinh\big(\sqrt{-K}\,v_k(x)\big)=\frac{1}{\sqrt{-K}}\sinh\left(\operatorname{sign}(\alpha_k)\,\beta_k\,\left|\sinh^{-1}\left(\sqrt{-K}\,\frac{\alpha_k}{\beta_k}\right)\right|\right),
>    $$
>    and the full PLFC output lies on the hyperboloid:
>    $$
>    y =
>    \begin{bmatrix}
>    \sqrt{(-K)^{-1} + \|y_s\|_2^2} \\
>    y_s
>    \end{bmatrix}.
>    $$
>
> Comparing these four formulas makes the situation unambiguous:
>
> - HCNN and Hypformer reuse this LFC (Hypformer just adds a curvature-conversion factor, which collapses to HCNN’s LFC when curvature is fixed). So we have already compared with both two.
> - Our PLFC is built from intrinsic Lorentz point–to–hyperplane distances, not from an ambient Euclidean affine map \(Wx\), and thus is mathematically a different layer; **the different functional forms in the formulas above make it very clear that PLFC is totally different from the LFC.**
> - Experiments in the paper already show that PLFC is strictly stronger than the standard LFC baseline (HCNN), so the advantage is both theoretical and empirical than HCNN and Hypformer.

---

### Author Response · Authors · 2025-11-26
**General Response**

We sincerely thank Reviewers qw2r, 9oRv, 98q5, and UjqT for their constructive reviews and insightful feedback, which have helped us enhance the quality and clarity of our paper. All the changes described below have been incorporated into the revised manuscript.

---

## Positive Feedback from Reviewers

1. Proposes **a new point-to-hyperplane fully connected layer (PLFC)** that utilizes intrinsic Lorentzian distance for logits to match the data's latent hierarchy
2. The proposed point-to-hyperplane Lorentz fully connected layer is new in the literature of hyperbolic neural networks
3. The proposed Intrinsic Lorentz Neural Network is **well motivated as existing hyperbolic architectures remain partially intrinsic**
4. The paper is also generally **well-written and extensive experimental results** show the improvements of the proposed Intrinsic Lorentz Neural Network, especially on TEB and GUE datasets.
5. The experiments on vision and genomics tasks show **potential interdisciplinary applications of hyperbolic neural networks.**
6. The paper is clearly structured, with **detailed mathematical formulas and reasonable ablation studies.**
7. GyroLBN that re-centers and scales the output with **efficient statistics.**

---

## Reviewer Concerns and Our Responses

1. **Hierarchical structure of datasets**: We now explicitly describe the semantic and biological hierarchies in CIFAR-10/100, TEB, and GUE, and **report their hyperbolicity (low $δ_{rel}$)** in a dedicated table and dataset section. This clarifies why these benchmarks are well-suited to hyperbolic modeling and motivates our choice of evaluation tasks.

2. **Generality across datasets and modules**: To demonstrate extensibility, we plug PLFC into **the attention layers of Hypformer** and evaluate on three classical graph benchmarks (Airport, Cora, PubMed), **achieving consistent improvements over the original model**. These results are now included in the main text, showing that our intrinsic design transfers across architectures and domains, not only CNNs and genomics.

3. **Definition and benefit of “Intrinsic”**: We provide a formal definition of intrinsic Lorentz layers, which require all computations to be expressible purely via Lorentzian operations, and contrast PLFC with prior partially intrinsic LFC designs. In addition, we prove a simple theorem showing that PLFC exactly preserves pre-logit margins while the extrinsic LFC contracts them, giving **a clear theoretical explanation of the benefit of intrinsic design**.

We have also addressed the specific concerns raised by individual reviewers:

1. For reviewer qw2r about why hyperbolic methods are effective for addressing latent hierarchical structure, we clarified where the latent hierarchical structure arises in CIFAR and the genomics benchmarks, and explained why hyperbolic methods are particularly effective in this regime, supported by δ_rel statistics and prior work.

2. For reviewer 9oRv about applying the intrinsic approach to build other kinds of neural network layers, we extended PLFC to the attention components of Hypformer and showed consistent gains on larger, widely used graph datasets, to address applicability to other architectures.

3. For reviewer 98q5 about the use of the “gyro-” terminology in the Lorentz model, we following previous work, now explicitly show that the Lorentz model indeed admits a gyrovector-space structure in the appendix, making our use of GyroLBN mathematically consistent.

4. For reviewer UjqT about the use of LFC and whether PLFC contains Lorentz transformation, we clarified the exact implementation of the LFC baselines consistent with previous work, and we explained why PLFC, being a nonlinear point-to-hyperplane construction, does not encompass and does not need to encompass all Lorentz transformations.

---
We believe these revisions and additions comprehensively address the reviewers’ concerns, and further validate the effectiveness, generality, and mathematical soundness of PLFC, GyroLBN, and our Intrinsic Lorentz Neural Network. Thanks for all reviewers' efforts again and we welcome any further questions or discussions regarding our approach.

---

> ### Author Response · Authors · 2025-12-01
> **Summary of Revisions in the Updated Manuscript**
>
> We thank all reviewers for their constructive comments, which helped us improve the clarity and presentation of our work. In the revised manuscript, we added several new analyses and experiments (highlighted in <font color='blue'>blue</font>) that directly address the raised concerns. Below, we summarize the main additions.
>
> * **Generality across datasets and modules (Section 5.3)**: We extend PLFC to the attention mechanism and conduct additional experiments on standard hierarchical graph datasets, as reported in Section 5.3.
>
> * **Hierarchical structure and hyperbolicity of image and genomics datasets (Appendix F.1)**: We add a description of these two types of datasets and report their hyperbolicity in Appendix F.1.
>
> * **Detailed definition and analysis of “intrinsic” (Section 4.1 and Appendix D)**: We provide a detailed definition of the intrinsic property of PLFC and analyze why the previous LFC layer is only partially intrinsic.
>
> * **Theoretical advantage of PLFC over LFC (Appendix E.2)**: We demonstrate the theoretical superiority of PLFC over LFC in a setting with a single Lorentz FC layer followed by a Lorentz MLR.
>
> * **Gyrovector space in the Lorentz model (Appendix C)**: We explain, step by step, how the Lorentz model forms a gyrovector space in Appendix C.
>
> We hope these additions address the reviewers’ concerns and further clarify the advantages and practicality of our proposed methods.

---

### Meta-Review · Area_Chair_irPJ · 2026-01-08

**Summary:**

Hyperbolic geometry has long promised a more natural home for data organised by hierarchy, yet many neural architectures that claim to inhabit this space continue to commute back and forth through Euclidean shortcuts. This paper proposes Intrinsic Lorentz Neural Networks, an architecture that conducts all computation strictly within the Lorentz model, and sets out to show why such geometric discipline matters.

At the centre of the proposal is a point-to-hyperplane Lorentz fully connected layer (PLFC). Rather than relying on ambient linear maps and subsequent projections, the layer defines logits as signed Lorentzian distances to learned hyperplanes. Around this, the authors assemble a suite of intrinsic components, most notably GyroLBN, a normalization layer that combines the geometric alignment of gyro-based methods with the efficiency of closed-form Lorentzian statistics.

Several reviewers raised concerns regarding novelty relative to prior Lorentz FC layers, the definition and benefit of “intrinsic” Lorentz operations, and the suitability of the chosen datasets for demonstrating hierarchical structure. In the rebuttal and revised manuscript, the authors address these points comprehensively by 1) providing a formal geometric definition of intrinsic Lorentz layers, 2) explicitly differentiating PLFC from prior LFC designs at the level of mathematical formulation and theoretical properties (including a margin-preservation result), 3) documenting hierarchical structure and measured hyperbolicity of all datasets, and 4) adding new experiments on standard graph benchmarks showing that PLFC acts as a drop-in improvement across architectures. While not all reviewers explicitly acknowledged these updates post-rebuttal, the substantive technical concerns raised in the reviews appear to be directly resolved in the revision, and no outstanding correctness or novelty issues remain unaddressed.

Taken as a whole, the revised paper makes a careful, technically grounded case for intrinsic Lorentz design. Its contributions are less dramatic than transformative, but they are clear, reusable, and well defended. For a field still negotiating the meaning of “fully hyperbolic” learning, this is a useful and credible step forward.

**Reviewer Concerns:**

Several reviewers raised concerns regarding novelty relative to prior Lorentz FC layers, the definition and benefit of “intrinsic” Lorentz operations, and the suitability of the chosen datasets for demonstrating hierarchical structure. In the rebuttal and revised manuscript, the authors address these points comprehensively by 1) providing a formal geometric definition of intrinsic Lorentz layers, 2) explicitly differentiating PLFC from prior LFC designs at the level of mathematical formulation and theoretical properties (including a margin-preservation result), 3) documenting hierarchical structure and measured hyperbolicity of all datasets, and 4) adding new experiments on standard graph benchmarks showing that PLFC acts as a drop-in improvement across architectures. While not all reviewers explicitly acknowledged these updates post-rebuttal, the substantive technical concerns raised in the reviews appear to be directly resolved in the revision, and no outstanding correctness or novelty issues remain unaddressed.

**Reviewer Scores:**

Reviewer qw2r: likely increase by 2: novelty, intrinsic definition, and dataset hierarchy concerns were addressed with formal theory, explicit comparisons, and added experiments.

Reviewer 9oRv: likely increase by 1: added graph/attention experiments and clearer intrinsic justification strengthen generality claims.

Reviewer 98q5: likely increase by 2: detailed technical objections (definitions, gyro terminology, theory, baselines) were directly resolved in revision.

Reviewer UjqT: likely increase by 1: baseline clarification and cross-domain results address correctness and scope concerns.

---

### Decision · Program_Chairs · 2026-01-26

Accept (Poster)